# A Fast Visible Wavelength 3-D Radiative Transfer Model for Numerical Weather Prediction Visualization and Forward Modeling

Steven Albers[1], Stephen M. Saleeby[2], Sonia Kreidenweis[2], Qijing Bian[2], Peng Xian[3], Zoltan Toth[4], Ravan Ahmadov[5,4], Eric James[5,4], and Steven D. Miller[2]

1 - Spire Global, Inc., Boulder, CO
2 - Colorado State University, Ft. Collins, CO
3 - U.S. Naval Research Laboratory, Monterey, CA
4 - Global Systems Division, ESRL/OAR/NOAA,
5 - CIRES at Global Systems Division, ESRL/OAR/NOAA

**Abstract:**

Solar radiation is the ultimate source of energy flowing through the atmosphere fueling all atmospheric motions. The visible wavelength range of solar radiation represents a significant contribution to the Earth's energy budget and visible light is a vital indicator for the composition and thermodynamic processes of the atmosphere from the smallest weather to the largest climate scales. The accurate and fast description of light propagation in the atmosphere and its lower boundary environment is therefore of critical importance for the simulation and prediction of weather and climate.

Simulated Weather Imagery (SWIm) is a new, fast and physically based visible wavelength 3-dimensional radiative transfer model. Given the location and intensity of the sources of light (natural or artificial) and the composition (e.g., clear or turbid air with aerosols, liquid or ice clouds, and precipitating rain, snow, or ice hydrometeors) of the atmosphere, it describes the propagation of light and produces visually and physically realistic hemispheric or 360° spherical panoramic color images of the atmosphere and the underlying terrain from any specified vantage point either on or above the Earth's surface.

Applications of SWIm include the visualization of atmospheric and land surface conditions simulated or forecast by numerical weather or climate analysis and prediction systems for either scientific or lay audiences. Simulated SWim imagery can also be generated for and compared with observed camera images to (i) assess the fidelity, and (ii) improve the performance of numerical atmospheric and land surface models, as well as through the use of the latter in a data assimilation scheme, (iii) improve the estimate of the state of atmospheric and land surface initial conditions for situational awareness and NWP forecast initialization purposes.

## 1. Introduction and Motivation

Numerical Weather Prediction (NWP) modeling is a maturing technology for the monitoring and prediction of weather and climate conditions on a wide continuum of timescales (e.g., Kalnay 2003). In NWP models, the large scale variability of the atmosphere is represented via carefully chosen and geographically systematically laid out prognostic variables such as vertically stacked latitude/longitude grids of surface pressure, temperature, wind, humidity, suspended (clouds) and falling (precipitating) hydrometeors, aerosol, etc. Using differential equations, NWP models capture temporal relationships among the atmospheric variables, allowing for the projection of the state of the atmosphere into the future. Short range NWP forecasts (called "first guess") can then be combined with the latest observations of atmospheric conditions to estimate the instantaneous weather conditions at any point in time (called analyzed state, analysis, or forecast initial condition), using Data Assimilation methods (DA, e.g. Kalnay, 2003).

The initialization of forecasts (and thus DA) plays a critical role in NWP as the more complete the information the analysis state has about the atmosphere, the longer pursuant forecasts will retain skill (e.g. Toth and Buizza, 2018). Hence the desire for DA to exploit as many observations, and from as diverse a set of instruments as possible. Some observations are in the form of model variables, in which case, after temporal and/or spatial interpolations, they can be directly combined with a model first guess (i.e., "direct" measurements or observations). Many other instruments, however, observe quantities that are different but related to the model variables (i.e., "indirect" measurements).

Indirect observations in the form of visible wavelength light intensity such as those from high (down to 30 second time frequency and 500m pixel) resolution imagers aboard a family of geostationary satellites (e.g., Himawari, GOES-R Advanced Baseline Imager, ABI, Schmit et al., 2017), and from airborne or ground-based cameras offer unique opportunities. First, unlike most other observations, light intensity is readily convertible to color imagery, offering a visual representation of the environment to both specialized (researchers or forecasters) and lay (the general public) users. Note that by far, visual perception is humans' most informative sense. Secondly, high resolution color imagery provides a unique window into fine-scale land surface, aerosol, and cloud processes that are critical both for the monitoring and nowcasting of convective and other severe weather events, as well as for the assessment and refinement of modeled energy balance relationships crucial for climate forecasting.

Information on related processes derivable from currently available other types of observations is limited in spatiotemporal and other aspects compared to color imagery.

Physically, color imagery is a visual representation of the intensity of different wavelength light (i.e. spectral radiance) reaching a selected point (i.e., location of a photographic or imaging instrument) from an array of directions determined by the design of the instrument, at a given time. For computational efficiency, radiative processes are vastly simplified in NWP models and typically resolve (Sun to

atmospheric or land surface gridpoint) only how solar insolation, in a one dimensional manner, affects the temperature conditions in the atmosphere and on the land surface.

Color imagery clearly reflects (no pun intended) the geographical distribution and

physical characteristics of cloud, aerosol, and land surface conditions in the natural environment. Some of the quantities used in NWP models to represent such conditions include the amount of moisture, various forms of cloud forming and falling hydrometeors, the amount and type of aerosols, as well as the amount and type of vegetation and snow cover on the ground, and water surface wave characteristics.

Light processes recorded in color imagery constitute indirect measurements of such natural process that before their possible use in the initialization of NWP models, must be quantitatively connected with NWP model prognostic variables.

In the assimilation of direct observations, the value of model variables in the first

guess is adjusted toward that of observations (based on the expected level of error in each, e.g. Kalnay, 2003). In the first step of assimilating indirect observations, simple models (called "forward" models or operators) are used to create "synthetic" observations based on model variables. Synthetic observations simulate what measurements we would get had instruments been placed in a world consistent with

the abstract conditions of an NWP first guess forecast. The model-based synthetic observations then can be compared with real-world measurements of the same (non-model) quantities. Utilizing an adjoint, or ensemble-based inverse of the forward operator, or other minimization procedure, the first guess forecast variables are then adjusted to minimize the difference between the simulated and real

observations. In case of visible light measurements, observations can be considered to be in the form of color (or multi-spectral visible) imagery.

Beyond their expanding use in DA applications, the simulation of color imagery from model variables via forward operators has another important purpose: the

visualization of 4D NWP analysis and forecast fields. Visualization renders the complex NWP data laid out in 3 dimensions in space and one across model variables) readily perceptible by both expert and lay audiences, facilitating a unique validation and communication of analysis and forecast information.

This study is intended to introduce SWIm, and describe what has been done so far, and suggest a roadmap for the future. Section 2 is a brief review of the general properties and limitations of currently available multispectral radiance and color imagery forward operators. The main contribution of this paper is the introduction of the recently developed fast color imagery forward (or color visible radiation transfer) model called Simulated Weather Imagery (SWIm, Section 3). Section 4 explores two application areas for SWIm: the visualization and validation of NWP analysis and forecast fields, as well as a vision for the assimilation of color imagery observations into NWP analysis fields. Closing remarks and some discussion are offered in Section 5.

## 2. Color Imagery and Spectral Radiance Forward Modeling

Light observations used in multispectral visible imagery are affected by three main factors: (1) the light source (its location and intensity across the visible spectrum); (2) the medium through which the light travels (the composition and density of its constituents in 3D space); and (3) the location where the light is observed or perceived (Fig. 1). Conceptually, the modeling of how light from a given source propagates through a medium and affects an instrument or receptor involves a realistic (a) relative placement of the light source, medium, and receptor with respect to each other; (b) representation of light emission from its source; (c) description of the medium (from an NWP analysis of the atmosphere and its surroundings); (d) simulation of how light is modified as it travels through the medium via absorptive and diffusive processes; and (e) simulation of the response of the instrument or human observer to the natural stimuli. Full, end-to-end color imagery forward modeling involves the specification of (a) and (b), an estimation of (c), the simulation of processes described in (d) ("ray-tracing"), as well as the consideration of the impact of radiation (e).

Light propagation has been extensively studied from both experimental and theoretical perspectives. The scientifically most rigorous treatment involves the study of how individual photons are affected by, and a stochastic analysis of, the expected or net effect of scattering and absorption. Named after the stochastic concept involved, this line of inquiry and the related methodology is called the "Monte Carlo" approach. As noted in Table 1, a Monte Carlo approach (e.g., Mayer, 2009) works in a

wide variety of situations with a wide array of 3-D atmospheric fields, arbitrary vantage points, and day/night applications. The Monte Carlo is the only listed package the authors have seen that produces similar images with visually realistic colors as seen from the ground. Table 1 also lists the characteristics of some other widely used radiative transfer models. Whereas the Monte Carlo (MC) model is physically more rigorous, it is computationally much more intensive than some of the other methods. The computational efficiency of the other methods come at a cost of significant approximations or other limitations. For example, the Rapid Radiative Transfer Model (RRTM) provides irradiance at different grid levels and is used as a radiation parameterization package in NWP models. As typical for such packages, RRTM operates in single columns, hence it cannot produce 3-D directional imagery that the Monte Carlo approach can. The Community Radiative Transfer Model (CRTM, Kleespies et al., 2004) is used for both visualization and as a radiative forward operator in variational and related DA systems. The Spherical Discrete Harmonic Ordinate Method (SHDOM, Evans, 1998, Doicu et al., 2013) is another sophisticated radiative transfer model often used in fine scale research studies. SHDOM can produce imagery with good physical accuracy.

Table 1 also lists the characteristics of SWIm, the recently developed method that the next section describes in some detail. SWIm was designed for the rapid production of color imagery under a wide range of conditions. To satisfy these requirements, approximations to the more rigorous treatment of some physical processes had to be made. The level of approximations was carefully chosen to improve computational efficiency without unnecessarily sacrificing accuracy. By considering human color vision perception, SWIm produces images that are visually realistic. This feature is used in other visualizations (e.g. Klinger et al, 2017) that use MYSTIC (Mayer, 2009), though to our knowledge isn't always considered for image display in the operational meteorology community. The color calculation allows the simulated images to be directly compared with photographic color images since it can accurately convert spectral radiance values into appropriate displayed RGB values on a computer monitor as described in Section 3.8. As discussed in the rest of this study, with these features, SWIm occupies a niche for the versatile visualization and validation of NWP analyses and forecasts, as well as for the assimilation of color imagery observations aimed at improved NWP initialization and nowcasting applications.

## 3. Ray Tracing Methodology

SWIm considers the Sun and the Moon (if it is sufficiently bright) as nearly point day- and night-time light sources. Information on the medium through which light travels is

obtained from 3-D NWP analysis and forecast hydrometeor and aerosol fields. To simulate the propagation of light, SWIm invokes an efficient simplified ray tracing approach that can be benchmarked against results from more sophisticated radiative transfer packages, including the Monte Carlo method. There are two main sets of rays that are traced for scattering and absorption calculations. The first is from the sun (forward direction, step 1a in Table 2) and the second is from the observer (backward direction, step 1b), making SWIm a forward-backward ray-tracing procedure (see Fig. 1). These traces are calculated over the model grid for the gas, aerosol, and hydrometeor components. Since the actual atmosphere extends above and if it is a limited area model (LAM), also laterally outside the model grid, an additional separate and faster ray-tracing step is done that considers just the gas and horizontally uniform aerosol components beyond the limited model domain. An algorithmic procedure then combines these results to arrive at the final radiance values and corresponding image display. The above steps are summarized in Table 2 below.

For gas and aerosols, we evaluate the optical depth, $\tau$, to determine transmittance $T$, where $T = \frac{I}{Io} = e^{-\tau}$. $\tau$ is the number of mean free paths. $Io$ is the initial intensity of the light beam and $I$ is the attenuated intensity. The extinction coefficient $\beta$ is integrated along the beam path to yield the optical depth:

$$\tau = \int \beta \, ds \tag{1}$$

where ds is a distance increment traveling along the light ray. The initial forward ray-tracing from the sun through the 3-D grid (Step 1a, shown as the yellow rays in Fig. 1) is tantamount for producing a 3-D short wave radiation field. For visually realistic color imagery generation, ray-tracing is done multi-spectrally at three reference wavelengths $\lambda$ corresponding to the primary colors of human vision and display devices: 615nm (Red), 546nm (Green) and 450nm (Blue). The specific wavelengths were chosen as a compromise between the locations of peaks in the Commission Internationale de l'Eclairage (CIE) color matching functions (Section 3.8) and a desire to have more uniformly spaced wavelengths that give independent samples of the visual (and solar) spectrum. The calculated radiances are scaled to the solar spectral radiance at the top of the atmosphere.

### 3.1 Solar irradiance and radiance

The top of atmosphere (TOA) solar irradiance $E_{TOA}$ at normal incidence (sun located at zenith) is assumed to be $\frac{1362 \, W/m^2}{r^2}$ where $r$ is the Sun-Earth distance in astronomical

units. This TOA irradiance can be expressed in terms of spectral irradiance $E_{TOA,\lambda}$ by considering the solar spectrum in units of W/m²/nm. We can consider the SWIm image output in the form of spectral radiance $L_\lambda$ in the spherical image space. $L_\lambda$ corresponds to surface brightness and customarily is represented in units of W/m²/sr/nm. For numerical convenience the spectral radiance can be normalized to be in solar relative units based on the TOA solar spectral irradiance, distributed (e.g. scattered) in a hypothetical uniform fashion over the spherical image space extending over a solid angle of $4\pi$ steradians. We will denote solar normalized (or relative) spectral radiance using the symbol $L_\lambda^{'}$. Thus

$$L_\lambda^{'} = \frac{4\pi L_\lambda}{E_{TOA,\lambda}} \quad . \tag{2}$$

It is interesting to note that sunlight reflected from a white Lambertian surface oriented normal to the sun has $L_\lambda^{'} = 4$.

Once we calculate SWIm spectral radiance values at each pixel it is possible to estimate the Global Horizontal Irradiance (GHI) by first integrating spectral radiance weighted by $cos(z)$ over the solid angle of the hemispherical sky to yield spectral irradiance. The GHI is typically calculated by integrating the spectral radiance from 300nm to 3000nm. However, SWIm only samples wavelengths within a narrower range from 400nm to 800nm. Despite this inconsistency, we can make an assumption when integrating over the wider spectrum that the resulting irradiance is nearly proportional to the spectral irradiance at the 546nm green wavelength used in SWIm calculations. This approximation is reasonably accurate in cases where the global irradiance has a similar spectrum to the incident solar radiation, as seen on a mostly cloud-free day in Fig. 2. For example the slight reddening of the direct solar radiation due to Rayleigh scattering is often partially compensated by the blue color of the sky that represents the diffuse irradiance. Overcast sky conditions should work as well as long as the sky is a relatively neutral gray color. Indeed, the existing algorithm generally provides a close match when comparing SWIm generated GHI values to actual GHI values measured with a pyranometer at the National Renewable Energy Laboratory (NREL) in Golden, CO, except it tends to overestimate the GHI in uniform overcast conditions. We are considering whether this is due to the radiative transfer assumptions in SWIm or an underestimation in the analyzed 3D hydrometeors and associated cloud optical thickness.

In a worst case scenario of a pure Rayleigh blue sky, we calculate that the normalized spectrum integrated from $0.3\,\mu$ to $3.0\,\mu$ has a crossover point at 530nm with the solar spectrum, yielding an irradiance underestimation of about 11% of the diffuse component

when a SWIm reference green wavelength of 546nm is used. With a high sun in a clear sky this reduces to about 1% total GHI error since the Rayleigh scattered diffuse component is a small proportion of the total irradiance. For this error estimation, we integrated the Planck function at 5800K to represent an approximate solar spectrum and compared this with the Planck function convolved with the $\lambda^{-4}$ intensity vs wavelength associated with Rayleigh scattering. The error be reduced by a more detailed consideration of the three SWIm reference wavelengths. A simple preliminary correction parameter based on atmospheric water vapor content has been added to account for absorption in the near-IR wavelengths. This presently neglects separate consideration of direct and diffuse solar irradiance.

## 3.2 Other light sources and atmospheric effects

With its realistic 3D ray tracing, SWIm is able to simulate a number of daytime, twilight, and nighttime atmospheric light effects, including consideration of a spherical atmosphere. This involves various light sources including moonlight, city lights, airglow, and astronomical objects. These will be demonstrated in a separate paper.

## 3.3 Clear sky ray-tracing

To cover the full extent of atmosphere beyond the NWP model domain, a "clear sky" ray-tracing (Step 2) is conducted on a coarser angular grid compared with Step 1. The primary purpose of Step 2 is to provide a more direct account of the radiance produced by Rayleigh single scattering. A second purpose is to model the effect of aerosols that may extend beyond the top of the model grid, specified via a 1-D stratospheric variable. The accuracy of radiative processes associated both with stratospheric aerosols and twilight benefit from the vertical extent considered in this step, all the way up to about 100km. To calculate the solar relative spectral radiance, the ray-tracing algorithm integrates along each line of sight from the observer as

$$L'_{\lambda,clear} = P(\theta) \int e^{-\tau_s} e^{-\tau_o} d\tau_o \qquad (3)$$

where $\theta$ is the scattering angle shown in Fig. 1 and $P(\theta)$ is the phase function (described in section 3.4.1). $\tau_s$ is the optical thickness along the forward ray (yellow lines in Fig. 1) between the light source and each point of scattering and $\tau_o$ is the optical thickness along the backward ray (purple lines in Fig. 1) between the observer and each scattering point. We will denote this to be the clear sky radiance, that includes

the molecular component through the full atmospheric depth and aerosols above the model grid top.

### 3.4 Hydrometeors

As the light rays are traced through the model grid (yellow rays in Fig. 1, Step 1a in Table 2) their attenuation and forward scattering is determined by considering the optical thickness of intervening clouds and aerosols along their paths. The optical thickness between each 3D grid point and the light source $\tau_s$ is calculated. An estimate of back-scatter fraction $b$ is incorporated to help determine the scalar irradiance $E_\lambda$ (direct + scattered) at a particular model grid point. $b$ is assigned a value of $.063$ for cloud liquid and rain, $.14$ for cloud ice and snow, and $.125$ for aerosols. Scalar irradiance is the total energy per unit area impinging on a small spherical detector. Based on a cloud radiative transfer parameterization (Stephens, 1978), a simplified version was developed for each 3D grid point as follows,

$$T_1 = 1 - \frac{b\tau_s}{(1+b\tau_s)} \tag{4}$$

where $T_1$ is the transmittance of a cloud assuming light rays are scattered primarily along a straight line from light source to grid point. We define auxiliary eq. 5 that assumes some light rays can have multiple scattering events that travel predominantly perpendicular to an assumed horizontal cloud layer and $z_0$ is the solar zenith angle. This allows for cases with a vertical cloud thickness significantly less than horizontal extent, and the multiply scattered light will largely travel in an envelope that curves on its way from the light source to the observer.

$$T_2 = 1 - \frac{b\tau_s \cos z_0}{(1 + b\tau_s \cos z_0)} \tag{5}$$

Eq. 6 is used on the assumption that the overall transmittance $T$ will depend on the dominant mode of multiple scattering between source and observer, either along a straight line $T_1$, or the light scatters mostly perpendicular to the cloud layer $T_2$, allowing a shorter path to travel through the hydrometeors.

$$T = max(T_1, T_2 \cos z_0) \tag{6}$$

Considering the direct irradiance component, the hydrometeor extinction coefficient is largely dependent on the effective radius of the cloud hydrometeor size distribution. The expression in eq. 7 is adapted from (Stephens, 1978).

$$\beta = \frac{1.5\,CWC}{r_e \varrho_h} \qquad (7)$$

$\beta$ is the extinction coefficient used when we integrate along the light ray from the light source the grid point to calculate $\tau_s$, $CWC$ is the condensed water content, $r_e$ is the effective radius, and $\varrho_h$ is the hydrometeor density based on the hydrometeor type and the effective radius -- all defined at the current model grid point.

The effective radius is specified based on hydrometeor type and (for cloud liquid and ice) $CWC$. For cloud liquid and cloud ice, larger values of $CWC$ translate to having larger $r_e$ and smaller $\beta$. In other words larger hydrometeors have a smaller area to volume ratio and scatter less light per unit mass. When we trace light rays through a particular grid box, the values of CWC are trilinearly interpolated to help prevent rectangular prism shaped artifacts from appearing in the images.

We can now write eq. 8 for the scalar irradiance at the grid point, here assuming the surface albedo to be $0$,

$$E_{x,y,z,\lambda} \;=\; e^{-\tau_R}\; T\, E_{TOA,\lambda} \qquad (8)$$

where $\tau_R$ represents the optical thickness of the air molecules between the light source and observer that engage in Rayleigh scattering. Light reflected from the surface or scattered by air molecules and reaching the grid point are neglected here and considered in subsequent processing.

### 3.4.1 Single Scattering

The single scattering phase function has a sharp peak near the sun (i.e. forward scatter) that generally becomes stronger in magnitude for larger hydrometeors. Cloud ice and snow also have sharper forward peaks than liquid, particularly for pristine ice. A linear combination of Henyey-Greenstein (HG) functions (Henyey and Greenstein, 1941) is employed to specify the angularly dependent scattering behavior (phase function) for each hydrometeor type, producing curves shown in Figure 3. Linear combinations employing several of these functions are used as a simple way to reasonably fit the angular dependence produced by Mie scattering. If more detailed size distributions (and

particle shapes for ice) are available, a more exact representation of Mie scattering can be considered through the use of Legendre polynomial coefficients and a lookup table, or through other parameterizations (e.g. Key et al., 2002). Given the values of asymmetry factor $g$, the individual Henyey-Greenstein terms (6) are combined and normalized to integrate to a value of $4\pi$ over the sphere, so that their average value is 1, thus conserving energy. $\theta$ is the scattering angle (Fig. 1), and $i$ represents an individual HG phase function term that is linearly combined to yield the overall phase function. Specific values of $f_i$ and $g_i$ are given in expressions for $P_{thin}(\theta,\lambda)$ in section 3.4.2 and in Appendix B. These provide for light scattered in both forward and backward directions.

$$p_i(\theta, g_i) \;=\; \frac{1 - g_i^2}{[1 + g_i^2 - 2g_i\,cos(\theta)]^{3/2}} \tag{9}$$

The overall phase function is given by

$$P(\theta) \;=\; \sum_i f_i\, p_i(\theta, g_i) \;, \tag{10}$$

noting that $\sum_i f_i \;=\; 1$. When $\tau_o \ll 1$ we can use a thin atmosphere approximation to estimate the solar relative spectral radiance due to single scattering.

$$L_\lambda' \simeq P(\theta)\, \tau_o \omega \tag{11}$$

This relationship applies to hydrometeors as well as aerosols and the molecular atmosphere. In practice the ray tracing algorithm considers extinction between the sun and the scattering surface as well as between the scattering surface and the observer, thus eq. 11 applies given also that $\tau_s \ll 1$ along the ray traced from the observer. $\omega$ is the single scattering albedo as discussed below in Section 3.5.1. To allow a more general handling of larger values of $\tau_s$ a more complete formulation of the solar relative radiance is as follows:

$$L_\lambda' \;=\; P(\theta) \int_{\tau_o=0}^{2} E\, e^{-\tau_o}\, d\tau_o \tag{12}$$

### 3.4.2 Multiple scattering

When the optical thickness along the forward or backward paths approaches or exceeds unity, contributions to the observed signal from multiple scattering events

become too significant to approximate via single-scattering. A rigorous, though time-consuming approach such as Monte-Carlo would consider each scattering event explicitly. Instead, here we use a more efficient approximation that arrives at a single scattering phase function that approximates the bulk effect of the multiple scattering events. Several terms that interpolate between optically thin and thick clouds are used as input for this parameterization as described below.

Thick clouds seen from near ground level can be either directly or indirectly illuminated by the light source. As illustrated by the light rays in Figure 1, direct illumination corresponds to $lim_{\tau_o \to 0} \tau_s = 0$. A fully lit cloud surface will by definition have no intervening material between it and the sun. Conversely, indirect illumination implies that $lim_{\tau_o \to 0} \tau_s \gg 0$. The indirect illumination case is assumed to have anisotropic brightness that is dependent on the upward viewing zenith angle $z$ of each image pixel. This modulates the transmitted irradiance value associated with the point where this light ray intersects the cloud. Note that when looking near the horizon, the multiple scattering events have a higher probability of having at least one surface reflection, resulting in an increased probability of photon absorption. Under conditions of heavily overcast sky and low surface albedo, this results in a pattern of a darker sky near the horizon and a steadily brightening sky toward the zenith. Such a pattern typically seen in corresponding camera images is reasonably reproduced with the use of a normalized brightness given by $\frac{1 + 4\,cos(z)}{3}$. The direct illumination case is similar except that the irradiance value is given by the solar irradiance and the relative brightness depends on the scattering angle, peaking in the antisolar direction.

Intermediate values of $\tau_o$ are given empirical phase functions with decreasing effective values of $g$ as $\tau_o$ increases, similar to the concepts described in Piskozub and McKee, 2011. As $\tau_o$ increases with thicker clouds, the scattering order also increases and the effective phase function becomes flatter. When $\tau_o > 1$, we consider an effective asymmetry parameter $g' = g^{\tau_o}$, where $g$ is the asymmetry parameter term used for single scattering. The strategy of using $g'$ in the manner shown below underscores the convenience of using HG functions in the single scattering phase function formulation. $g'$ is combined with additional empirical functions that help give simulated cloud images that are similar to observed clouds of varying optical thicknesses. The goal is to have the solar aureole gradually expanding with progressively thicker clouds, eventually becoming diluted into a more uniform cloud appearance. In the case of cloud liquid, looking at a relatively dark cloud base where $\tau_s \gg 1$, we arrive at this semi-empirical formulation for the effective phase function.

$$P(\theta,\lambda,z) = c_1 P_{thin}(\theta,\lambda) + c_2 P_{thick}(\theta,z) , \tag{13}$$

where $c_1$ and $c_2$ are weighting coefficients.

$$c_1 = e^{-(\tau_o/10)^2} \frac{E_\lambda}{E_{TOA,\lambda}} \quad \text{and} \tag{14}$$

$$c_2 = 1 - c_1 . \tag{15}$$

Given the empirical nature of this formulation, $c_1 + c_2$ isn't constrained to equal $1$. For optically thin clouds we calculate $P_{thin}$ considering the three reference wavelengths $\lambda$ introduced in section 3 and associated asymmetry parameters $g_\lambda$:

$$g_\lambda = (0.945, 0.950, 0.955) \tag{16}$$

$$f_1 = 0.8 \times \tau_o \tag{17}$$

$$P_{thin}(\theta,\lambda) = f_1 \, p(\theta,g_\lambda^{\tau_o}) + (1.06 - f_1)\, p(\theta,0.6^{\tau_o}) + 0.02\, p(\theta,-0.6) - 0.08\, p(\theta,0) \tag{18}$$

$P_{thick,h}$ represents the effective phase function of a directly illuminated (high radiance) optically thick cloud, typically the sunlit side of a cumulus cloud. We represent such clouds as sections of spherical surfaces with a surface brightness varying as a function of $\theta$.

Our neighboring planet Venus offers an astronomical example for the radiative behavior of such a cloudy spherical surface. For the planet as a whole, Venus has a well established phase function $\triangle m$ (in astronomical magnitudes, Mallama et al., 2006). Changes in the average radiance of the illuminated portion of the sphere can be approximated by dividing the total brightness (numerator of eq. 19) by the illuminated fractional area. This denominator is based on its current illuminated phase (or equivalently the scattering angle $\theta$).

$$P_{thick,h}(\theta) = \frac{(1.94 / 10^{(0.4 \times \triangle m(\theta))})}{(1 - cos(\theta)) / 2} \tag{19}$$

The effective phase function of an indirectly illuminated thick low irradiance cloud (e.g., a dark cloud base, $P_{thick,l}$) can be written as:

$$P_{thick,l}(z) = \frac{1 + 2\,cos(z)}{3} \tag{20}$$

We combine the high irradiance and low irradiance cases for thick clouds depending on the irradiance of the surface of the cloud facing the observer, such that

$$P_{thick}(\theta,z) = 2\,c_3\,P_{thick,h}(\theta) + 4\,c_4\,P_{thick,l}(z)\,. \tag{21}$$

$c_3$ and $c_4$ are further weighting coefficients blending the component phase functions such that

$$c_3 = e^{-(\tau_o/10)^2}\,\frac{E_\lambda}{E_{TOA,\lambda}} \qquad and \tag{22}$$

$$c_4 = 1 - c_3\,. \tag{23}$$

The coefficients were experimentally determined by comparing simulated images of the solar aureole from clouds having various thicknesses, with both camera images and visual observations. Similarly constructed effective phase functions are utilized for cloud ice, rain, and snow (Appendix B).

### 3.4.3 Cloud Layers Seen from Above

As a simple illustration for cases looking from above we consider a homogeneous cloud of hydrometeors having optical thickness $\tau$, being illuminated with the sun at the zenith (i.e. $z_o = 0$). The cloud albedo (assuming a dark land surface) can be parameterized as:

$$a = \frac{b\tau}{(1+b\tau)} \tag{24}$$

where b is the backscatter fraction (Stephens, 1978). $\tau$ here is considered to be along the slant path of the light rays coming from the sun ($\tau_s$ in Fig. 1). For values of $\tau \leq 1$, we can assume single scattering and $a \sim b\tau$, while for large $\tau$, $a > 0.9$ and asymptotes to just below 1.0 (not reaching 1.0 identically due to the presence of a very small absorption component term). We set $b$ based on a weighted average of the contribution to $\tau$ along the line of sight for the set hydrometeor types. Cloud liquid and rain use $b = .06$, cloud ice and snow use $b = .14$. Graupel has yet to be tested in SWIm, though we anticipate using $b = .30$.

For $\tau \gg 1$ (asymptotic limit) the cloud albedo $a$ can be translated into an approximate reflectance value through a division by $\mu_o$, where $\mu_o = cos\ z_o$. This is the case since thick cloud (or aerosol) layers act approximately as Lambertian reflectors (with $g \to 0$) for the high order scattering component (Piskozub and McKee, 2011, Gao et al., 2013, Bouthers et al., 2008). When a given photon is scattered many times, the stochastic nature of the scattering causes the correlation between the direction of propagation of the photon and the direction of incident radiation to greatly decrease. To improve the accuracy we address the anisotropies that occur using a bidirectional reflectance distribution function (BRDF) as specified with a simple formula for the anisotropic reflectance factor (ARF).

$$ARF = \frac{b_1 + b_2\ cos(z)\ cos(z_0) + P(\theta, g, f_b)}{4\ cos(z)\ cos(z_0)} \tag{25}$$

$z$ is the zenith angle of the observer as seen from the cloud. A DHG phase function (eq. 27) is used as a simple approximation for an assumed water cloud where $g = 0.7$ and $f_b = 0.4$. This parameterization (Kokhanovsky, 2004) using $b_1 = 1.48$ and $b_2 = 7.76$ produces results consistent with graphical plots depicting the ARF for selected solar zenith angles (Lubin and Weber, 1994). When all orders of scattering are considered, the ARF remains close to 1 when the zenith angles $z, z_o$ are small. A large solar zenith angle shows preferential forward scattering causing the ARF to increase markedly with low scattering angles. Even with this enhancement, inspection of ABI satellite imagery suggests the reflectance factor, $\mu_o \times ARF$, generally stays below 1.0 in forward scattering cases.

In cases where $\tau < 1$ we are in a single scattering (or low-order) regime and the dependence of reflectance on $\mu_o$ goes away. In practice, this means that thicker aerosol (or cloud) layers will generally decrease in reflectance with a large $z_o$, while the reflectance holds more constant for very thin layers (assuming molecular scattering by the gas component is small). This causes the relative brightness of thin aerosol layers, compared with thicker clouds and the land surface to increase near the terminator. Linear interpolation with respect to cloud albedo is used to arrive at an expression for solar relative radiance taking into account the low $\tau$ and high $\tau$ regimes.

$$L_\lambda' = P(\theta)\ (1 - a) + ARF\ a \tag{26}$$

Here $P(\theta)$ is specified in Eq. 13. It should be noted that absorption within thick clouds has yet to be included in specifying the cloud albedo.

## 3.5 Aerosols

There are two general methods for working with aerosols in SWIm. The first uses a 1-D specification of the aerosol field that runs somewhat faster than a 3-D treatment. The second, newer, approach considers the 3-D aerosol distribution described in detail herein. Aerosols are specified by a chemistry model in the form of a 3-D extinction coefficient field. Various optical properties are assigned based on the predominant type (species) of aerosols present in the model domain.

## 3.5.1 Single Scattering

To determine the scattering phase function clouds and aerosols are considered together and aerosols are simply considered as another species of hydrometeors. For a case of aerosols only, the phase function $P(\theta)$ is defined depending on the type of aerosol. The Double Henyey-Greenstein (DHG, eq. 27) function (Louedec et al., 2012) is the basis of what is used to fit the phase function.

$$P(\theta,g,f_b) = (1 - g^2) \left[ \frac{1}{1+g^2 - 2g\ cos(\theta)} + f_b \left( \frac{3\ cos^2(\theta) - 1}{2\ (1+g^2)^{3/2}} \right) \right] \quad (27)$$

This function has the property of integrating to 1 over the sphere representing all possible light ray directions - $\theta$ is the scattering angle, and the asymmetry factor $g$ represents the strength of the forward scattering lobe. The weaker lobe in the back scattering direction is controlled by $f_b$.

Dust generally has a bimodal size distribution of relatively large particles. Accounting for both the coarse and fine mode aerosols, and for fitting the forward scattering peak, a linear combination of a pair of DHGs (eq. 11) can be set by substituting $g_1$ and $g_2$ for $g$. As an example we can assign $g_1 = .962$, $g_2 = .50$, $f_b = .55$, $f_c = .06$, where $f_b$ is the term for the backscatter peak and $f_c$ is the fraction of photons assigned to the first DHG using $g_1$:

$$P(\theta,g_1,g_2,f_b) = f_c P(\theta,g_1,f_b) + (1 - f_c)\ P(\theta,g_2,f_b) \quad (28)$$

Smoke and haze are composed of finer particles. Here we can also specify a combination of $g_1$, $g_2$, and $f_c$ to help in fitting the phase function. The asymmetry factor values of $g$, $g_1$ and $g_2$ each have a slight spectral variation to account for the variation in size parameter with wavelength. This means that a slight concentration of

bluer light occurs closer to the sun or moon. The overall asymmetry factor $g$ is related to the component factors $g_1$ and $g_2$ as follows:

$$g = f_c g_1 + (1 - f_c) g_2 \qquad (29)$$

$g_1$ and $g_2$ are allowed to vary slightly between the three reference wavelengths (Section 3). In addition, each application of the DHG function uses an extinction coefficient that varies according to an Angstrom exponent, that in turn depends on $g$ at 546nm. This allows for the spectral dependence of extinction. Coarser aerosols will have a higher asymmetry factor (i.e. a stronger forward scattering lobe), a lower Angstrom exponent and a more uniform extinction at various wavelengths giving a more neutral color. The value of $f_c$ can be set to reflect contributions from a mixture of aerosol species. We can thus specify the aerosol phase function with four parameters $g_1$ , $g_2$, $f_c$, and $f_b$ .

The single scattering albedo $\omega$ can also be specified for each wavelength to specify the fraction of attenuated light that gets scattered. $\omega$ represents the probability that a photon hitting an aerosol particle is scattered rather than absorbed, thus darker aerosols have $\omega$ significantly less than 1. The spectral dependence of $\omega$ is most readily apparent in the color of the aerosols as seen with back scattering. This applies either to a surface view opposite the sun, or to a view from above (e.g. space). Taking the example of hematite dust, the single scattering albedo $\omega$ is set to 0.935, 0.92, and 0.86 for our Red/Green/Blue reference wavelengths, respectively. This can eventually interface with a library of optical properties for a variety of aerosol types.

### 3.5.2 Optical Properties Assignment

In its current configuration, aerosol optical properties for the entire domain are assumed to be characterized by a single set of parameters in SWIm, reflecting the behavior of a predominant type or mixture of aerosols. The first row in Table 3 was arrived at semi-empirically for relatively dusty days in Boulder, CO, by setting values of the parameters and comparing the appearance of the solar aureole and overall pattern of sky radiance between simulated and camera images as well as visual observations.

The cameras being used aren't radiometrically calibrated, though we can approximately adjust the camera color and contrast on the basis of the Rayleigh scattering radiance distribution far from the sun on relatively clear days. We are thus limited to looking principally at relative brightness changes in a semi-empirical manner. The cameras

aren't using shadow bands, and generally have saturation due to direct sunlight within ~5-10 degrees radius from the sun. In some cases we supplement the cameras with visual observations (e.g. standing behind the shadow of a building) to assess the innermost portions of the aureole.

These days feature a relatively condensed aureole around the sun indicative of a contribution by large dust particles to a bimodal aerosol size distribution. This type of distribution has often been observed in AERONET (Holben et al., 1998) retrievals. The single scattering albedo is set with increased blue absorption as might be expected for dust containing a hematite component.

The second case of mixed dust and pollution was derived from AERONET observations over Saudi Arabia, calculating the phase function using Mie scattering theory (Appendix A), then applying a curve fitting procedure to yield the four phase function parameters described previously. In this case the single scattering albedo is spectrally independent. Simulated images for these two sets of phase function parameters are shown in Fig. 4.

### 3.5.3 Multiple Scattering

As with meteorological clouds, when the aerosol optical thickness along the forward or backward ray paths (Fig 1) approaches or exceeds unity, the contributions from multiple scattering increase. In a manner similar to cloud multiple scattering, we utilize a more efficient approximation that determines a single scattering phase function that is equivalent to the net effect of the multiple scattering events.

### 3.5.4 Aerosol Layers Seen from Above

Non-absorbing aerosols seen from above can be treated in a similar manner to cloud layers as described above (eq. 9). We now extend this treatment to address absorbing aerosols. SWIm was tested using 3D aerosol fields from two chemistry models running at Colorado State University (CSU): the Regional Atmospheric Modeling System (RAMS, Miller et al., 2019; Bukowski et al., 2019) and the Weather Research and Forecasting Model (WRF, Skamarock et al., 2008). SWIm was also tested with two additional chemistry models, the High Resolution Rapid Refresh (HRRR)-Smoke (Fig 5, available at https://rapidrefresh.noaa.gov/hrrr/HRRRsmoke) and the Navy Global Environmental Model (NAVGEM - Fig 6, Hogan et al., 2014). These tests yielded valuable information about how multiple scattering in absorbing aerosol layers can be handled.

For partially absorbing aerosols such as smoke containing black carbon or dust, in a thin layer we can multiply eq. (6) by $\omega$, the single scattering albedo to get the aerosol layer albedo.

$$a = \omega \, \frac{b\tau}{(1+b\tau)} \tag{30}$$

A more challenging case to parameterize is when $\tau \gg 1$ and multiple scattering is occurring. Each extinction event where a photon encounters an aerosol particle now also has a non-zero probability of absorption occurring. Here we can consider a probability distribution for the number of scattering events for each photon that would have been received by the observer if the aerosols were non-absorbing (e.g. sea salt where $\omega \sim 1$). We can define a new quantity $\omega'$ to represent a multiple scattering albedo.

$$a = \omega' \frac{b\tau}{(1+b\tau)} \tag{31}$$

For typical smoke or dust conditions $a$ will approach an asymptotic value between about 0.3 to 0.5. We plan to check the consistency of SWIm assumptions with previous work in this area such as in (Bartkey, 1968). Once the albedo is determined a phase function is used for thin aerosol scattering and a BRDF is used for thick aerosols. This is similar to the way that clouds are handled.

### 3.6 Combined clear sky and aerosol/cloud radiances

The clear sky radiance $L'_{\lambda,clear}$ is calculated through the whole atmosphere in Step 2, while the aerosol and cloud radiances (grouped into $L'_{\lambda,cloud}$) are determined within the more restricted volume of the model grid (Step 1b). As a post-processing step these quantities are merged together with this empirical procedure to provide the combined radiance $L'_{\lambda}$ at each location in the scene from the observer's vantage point. We define a quantity $f_{front}$ to be the conditional probability that a backward traced light ray from the observer is scattered or absorbed by the molecular component vs. being scattered or absorbed from the molecular component, aerosols, or hydrometeors. $\tau_1$ is denoted as the optical thickness of the molecular and aerosol component between the observer and where $\tau_o = 1$ ($\tau_o$ also having hydrometeors included). We then calculate the following:

$$f_{clear} = f_{front} + (1 - f_{front})(1 - e^{-\tau_o}) \tag{32}$$

$$f_{cloud} = (1 - e^{-\tau_o})\, e^{-\tau_1} \tag{33}$$

$$L'_\lambda = f_{clear}\, L'_{\lambda,clear} + f_{cloud}\, L'_{\lambda,cloud} \tag{34}$$

The above strategy permits the addition of blue sky from Rayleigh scattering in front of a cloud, based on the limited amount of atmosphere between observer and cloud.

**3.7 Land Surface**

When a backward-traced ray starting at the observer intersects the land surface we consider the incident and reflected light upon the surface that contributes to the observed light intensity, as attenuated by the intervening gas, aerosol, and cloud elements. Terrain elevation data on the NWP model grid is used to help determine where light rays may intersect the terrain. The land spectral albedo is obtained at 500m resolution using the Blue Marble Next Generation Imagery (BMNG, Stockli et al., 2005). The BMNG image RGB values are functionally related to spectral albedo for three Moderate Resolution Imaging Spectroradiometer (MODIS) visible wavelength channels. A spectral interpolation is performed to translate the BMNG / MODIS albedos into the three reference wavelengths used in SWIm.

For higher resolution display over the continental United States, an aerial photography dataset obtained from the United States Department of Agriculture (USDA) can also be used (Figs 7, 8). The associated National Agriculture Imagery Program (NAIP) data are available at 70cm resolution and is added to the visualization at sub-grid scales with respect to the model Cartesian grid. This dataset is only roughly controlled for spectral albedo, though it can be a good tradeoff with its very high spatial resolution.

To obtain the reflected surface radiance in each of the three reference wavelengths, we utilize clear-sky estimates of direct and diffuse incident solar irradiance. For the direct irradiance component, spectral albedo is converted to reflectance using the anisotropic reflectance factor $ARF$ that depends on the viewing geometry and land surface type. Thus reflectance $\varrho$ is defined as: $\varrho = a\,(ARF)$, where $a$ is the terrain albedo. The solar relative spectral radiance of the land surface is calculated as

$$L'_\lambda = \frac{4\,\varrho\,E_{\lambda H}}{E_\lambda} \tag{35}$$

where $E_{\lambda H}$ is the global horizontal spectral irradiance. This relationship can also be used for the diffuse irradiance component if we assign $ARF = 1$.

 Relatively simple analytical functions for $ARF$ are used over land with maximum values in the backscattering direction. Modified values of surface albedo and $ARF$ are used in the presence of snow or ice cover with maximum values in the forward scattering direction. A sun glint model with a fixed value of mean wave slope is used over water similar to earlier work (Cox and Munk, 1954), except that waves are given a random orientation without a preferred direction. Scattering from below the water surface is also considered. In the future, wave slope will be derived from NWP ocean wave and wind forecasts.

## 3.8 Translation into displayable color image

As explained earlier, spectral radiances are computed for three narrowband wavelengths, using solar-relative intensity units to yield a scaled spectral reflectance. This allows some flexibility for outputting spectral radiances, spectral reflectance, or more visually realistic imagery that accounts for details in human color vision and computer monitor characteristics. To accomplish the latter it is necessary to estimate spectral radiance over the full visible spectrum using the partial information from the selected narrowband wavelengths we have so far. Having a full spectrum is important when computing an accurate human color vision response (Bell et al., 2006). The procedure is to first perform a polynomial interpolation and extrapolation of the three narrowband (solar relative) reflectance values, then multiply this by the solar spectrum, yielding spectral radiance over the entire visible spectrum at each pixel location. The observed solar spectrum interpolated in 20nm steps is used for purposes of subsequent numerical integration.

Digital RGB color images are created by calculating the image count values with three additional steps:

1) Convolve the spectral radiance (produced by the step described in the above paragraph) with the CIE tristimulus color matching response functions to account for color perception under assumptions of normal human photopic vision. Each pixel of the image now specifies the perceived color in the XYZ color space (Smith and Gould, 1931). In this color system the chromaticity (related to color hue and saturation) is represented by normalized xy values and the perceived brightness is the Y value. The

normalization of the XYZ values to yield chromaticity specifies that x+y+z=1. The xyz chromaticity values represent the normalized perception for each of the three primary colors. An example illustrating the benefits of this procedure is the blue appearance of the daytime sky. We calculate a pure Rayleigh blue sky to have chromaticity values of x=.235, y=235. The violet component of the light is actually stronger than blue, but has less impact on the perceived color since we are less sensitive to light at that wavelength.

2) Apply a 3x3 transfer matrix that puts the XYZ image into the RGB color space of the display monitor.

$$
\begin{bmatrix} r \\ g \\ b \end{bmatrix} = \begin{bmatrix} 3.1894 & -1.5755 & -.4948 \\ -.9735 & 1.8951 & 0.0376 \\ 0.0635 & -.2160 & 1.2244 \end{bmatrix} \begin{bmatrix} X \\ Y \\ Z \end{bmatrix}
$$

(36)

This is needed in part because the colors of the display system are not spectrally pure. Another consideration is the example of spectrally pure violet light, perceived in a manner similar to purple (a mix of blue and red for those with typical trichromatic color vision). Violet is beyond the wavelengths that the blue phosphors in a monitor can show, so a small component of red light is mixed in to yield the same perception, analogous to what our eye-brain combination does. We make the assumption that the sun (the main source of illumination) is a pure white color as is very nearly the case when seen from space thus setting the white point to 5780K, the sun's approximate color temperature. Correspondingly, when viewing SWIm simulated color images, we also recommend setting one's display (e.g. computer monitor) color temperature to 5780K.

3) Include a gamma (approximate power law) correction with a value of 2.2 to match the non-linear monitor brightness scaling. With this correction the displayed image brightness will be directly proportional to the actual brightness of a scene in nature, giving realistic contrast and avoiding unrealistically saturated colors. With no correction, the contrast would be incorrect and the brightness off by an exponential amount.

Based on an extensive subjective assessment, this procedure gives a realistic color and contrast match if one looks at a laptop computer monitor held next to a scene in a natural setting on the ground, and is anticipated to perform well for air- and space-based simulations as well. The results have somewhat more subtle colors and contrast compared with many commonly seen Earth and sky images. The intent here is

to make the brightness of the displayed image proportional to the actual scene, and the perceived color to be the same as a human observer would see in a natural setting. This is without any exaggeration of color saturation sometimes occurring in satellite "natural color" image rendering (e.g. Miller et al., 2012) and even in everyday photography (subjective observation, Albers 2019). For example color saturation values of the sky in photography often exceeds the calculated values for even low aerosol conditions. A more complete consideration of the effects of atmospheric scattering and absorption in SWIm image rendering softens the appearance of the underlying landscape when viewed from space or otherwise afar. This is due to SWIm not suppressing the contribution of Rayleigh scattering to radiance as observed in nature.

## 4. Applications of SWIm

### 4.1 Model Visualization

The fast 3-D radiative transfer package called Simulated Weather Imagery has been developed to serve the development and application needs of high-resolution atmospheric modeling. Visually and physically realistic, full natural color (e.g., Miller et al., 2012) SWIm imagery, for example, offers a holistic display of numerical model output (analyses and forecasts). At a glance one can see critical weather elements such as the fields of clouds, precipitation, aerosols and land surface in a realistic and intuitive manner. Model results are thus more effectively communicated for interpretation, displaying weather phenomena that we see in the sky and contront in the surrounding environment. NWP information about current and forecast weather is readily conveyed in an easily perceivable visual form to both scientific and lay audiences.

The SWIm package has run on a variety of NWP modeling systems including the Local Analysis and Prediction System (LAPS, Toth et al., 2014), WRF, RAMS, HRRR (Benjamin et al., 2016), and NAVGEM. We can thus discern general characteristics of the respecting data assimilation and modeling systems including their handling of clouds, aerosols, and land surface (e.g. snow cover).

### 4.1.1 CSU RAMS Middle East Dust Case

Visualization of the RAMS model developed at CSU was done for a case featuring dust storms over the Arabian Peninsula and the neighboring region (Miller et al., 2019; Bukowski et al., 2019), as part of the Holistic Analysis of Aerosols in Littoral Environments Multidisciplinary University Research Initiative (HAALE-MURI). Figure 9 shows the result of this simulation from in-situ vantage points just offshore from Qatar in

the Persian Gulf at altitudes of 4km and 20m above sea level. With the higher vantage point we are above most of the atmospheric dust present in this case, so the sky looks bluer with the Rayleigh instead of Mie scattering being more dominant.

### 4.1.2 Other Modeling Systems

Figure 5 shows a space-based perspective of the December 2017 wildfires in Southern California using NWP data from the HRRR-Smoke system. Smoke plumes from fires and areas of inland snow cover are readily visible. SWIm has been most thoroughly tested with another NWP system called the Local Analysis and Prediction System (LAPS, Albers et al., 1996, Jiang et al., 2015). LAPS produces very rapid (5-minute) update and very high resolution (e.g. 500-m) analyses and forecasts of 3-D fields of cloud and hydrometeor variables. The LAPS cloud analysis is a largely sequential data insertion procedure that ingests satellite (including IR and 500-m resolution visible imagery, updated every 5-min), ground-based cloud cover and height reports, radar, and aircraft observations along with a first guess forecast. This scheme is being updated with a 3/4DVAR cloud analysis module that in the future will also be used in other fine scale data assimilation systems.

Figure 7 depicts a simulated panoramic view from the perspective of an airplane cockpit at 1km altitude using LAPS analysis with 500m horizontal resolution.  This is part of an animation designed to show how SWIm can be used in a flight simulator for aviation purposes. This visualization uses sub-grid scale terrain albedo derived from USDA 70cm resolution airborne photography acquired at a different time. SWIm has also been used to display LAPS-initialized WRF forecasts of severe convection (Jiang et al., 2015) showing a case with a tornadic supercell that produced a strong tornado striking Moore, Oklahoma in 2013.

### 4.2 Validation of NWP analyses and forecasts

Simulated images and animations from a variety of vantage points (on the ground, in the air, or in space, i.e. with multi-spectral visible satellite data) can be used by developers to assess and improve the performance of numerical model and data assimilation techniques. A subjective comparison of simulated imagery against actual camera images serves as a qualitative validation of both the model fields and the visualization package itself. If simulated imagery can well reproduce observed images under a representative range of weather and environmental conditions, this is an indication of the realism of the radiative transfer / visualization package (i.e., SWIm).

Discrepancies between simulated and observed images in other cases may be due to shortcomings in the analyzed or model forecast states.

Comparing analyses from LAPS with day-time and night-time camera images under cloudy, precipitating, and clear/polluted air conditions, SWIm was tested and can realistically reproduce various atmospheric phenomena (Albers and Toth, 2018). Since camera images are not yet used as observational input in LAPS, subjective and quantitative comparisons of high resolution observed and simulated weather imagery provides a valuable opportunity to assess the quality of cloud analyses and forecasts from various NWP systems, including LAPS, Gridded Statistical Interpolation (GSI, Kleist et al., 2009), HRRR, Finite Flow Following Icosahedral Model (FIM, Bleck et al, 2015), and the NAVGEM.

360° imagery, presented in either a polar or cylindrical projection, can show either analysis or forecast fields. Here, we present the results of ongoing developments of this simulated imagery, along with comparisons to actual camera images produced by a network of all-sky cameras that is located within our Colorado 500m resolution domain, as well as space-based imagery. These comparisons (summarized in Table 4) check the skill of the existing analysis of clouds and other fields (e.g. precipitation, aerosols, and land surface) at high-resolution.

**4.2.1 Ground-based observations**

Figure 10 shows a comparison between a simulated and a camera observed  all-sky image valid at the same time. The simulated image was derived from a 500m horizontal resolution, 5-min update cycle LAPS cloud analysis. Assuming realistic ray tracing and visualization, the comparison provides an independent validation of the analysis. In this case we see locations of features within a thin high cloud deck are reasonably well placed. Variations in simulated and observed cloud opacity (and optical thickness) are also reasonably well matched. This is evidenced by the intensity of the light scattering through the clouds relative to the surrounding blue sky, as well as the size (and shape) of the brighter aureole closely surrounding the sun. The brightness scaling being used for both images influences the apparent size of the inner bright (saturated) part of the solar aureole in the imagery. This saturation can occur either from forward scattering of the light by clouds and aerosols or from lens flare. The size also varies with cloud optical thickness and reaches a maximum angular radius at $\tau \sim 3$.

It is also possible to compare simulated and camera images to validate gridded fields of model aerosol variables. In particular, the effects of constituents other clouds, such as haze, smoke, or other dry aerosols on visibility under conditions analyzed or forecast by NWP systems can also be instantly seen in SWIm imagery (Albers and Toth 2018). Analogous to Fig. 10 (except its panoramic projection), Figure 11 shows a cloud-free sky comparison where aerosol loading was relatively high due to smoke. LAPS uses a simple 1-D aerosol analysis for a smoky day in Boulder, Colorado when the AOD was measured by a nearby AERONET station to be 0.7. The area within $\sim 5^{\circ}$ of the sun in the camera image should here be ignored due to lens flare.

Alternatively, solar irradiance computed by a solid angle integration of SWIm imagery has been compared (initially via case studies) with corresponding pyranometer measurements (Fig. 10). Qualitative comparison of the land surface state including snow cover and illumination can be compared with camera observations (not shown).

### 4.2.2 Space-based observations

For space-based satellite imagery, color images can be compared qualitatively and visible band reflectance can be used for quantitative comparisons.

Figure 12 shows observed imagery from the Earth Polychromatic Imaging Camera (EPIC) imagery aboard the Deep Space Climate Observatory (DSCOVR, Marshak et al, 2018) satellite, used as independent validation in a comparison with an image simulated by SWIm from a Global LAPS (G-LAPS) analysis. The DSCOVR imagery was empirically reduced in contrast to represent the same linear brightness (image gamma - Sec. 3.8) relationships used in SWIm processing. The LAPS analysis comprises 3-D hydrometeor fields (four species) at 21km resolution, in addition to other state and surface variables such as snow and ice cover. Visible and IR satellite imagery are utilized from GOES-16 and GOES-17, with first guess fields from a Global Forecast System (GFS) forecast, an operational model run by the National Oceanic and Atmospheric Administration (NOAA).

The horizontal location and relative brightness of the simulated vs. observed clouds match fairly closely in the comparison for many different cloud systems over the western hemisphere. The land surface spectral albedo also appears to be in good agreement, including areas of snow north of the Great Lakes. The sun glint model in SWIm shows the enhanced brightness surrounding the nominal specular reflection point in the ocean areas surrounding the Yucatan peninsula due to sunlight reflecting from waves assumed to have a normal slope distribution. This can help with evaluation of a coupled

wind and ocean wave model. There is some difference in feature contrast due to a combination of cloud hydrometeor analysis (e.g. the brightest clouds in central North America) and SWIm reflectance calculation errors, as well as uncertainty in the brightness scaling of the DSCOVR imagery, along with uncertainties in the snow albedo used in SWIm over vegetated terrain. The EPIC imagery shown was obtained from the displayed EPIC web products with color algorithms unknown to the authors, thus a better comparison could be performed using the radiance calibrated EPIC data, adjusted for Earth rotation offsets for the three color channels. The color image comparison is shown here to give an intuitive illustration of a multispectral comparison. The reflectance factor distribution for both SWIm and DSCOVR (now using the calibrated L1b radiance data) in a single channel (the red band) matches anticipated values from 5% in darkest clear oceanic areas to ~1.1 in bright tropical convection.

Figure 13 shows a comparison of color images over the Arabian peninsula and over the Persian Gulf as generated from MODIS Aqua observations and via SWIm simulation from a  RAMS model forecast. Various environmental conditions such as lofted dust (near the Arabian peninsula and over the Persian Gulf), liquid (low) and ice (high) clouds can be seen. The microphysics and chemistry formulations in the RAMS model can be assessed and improved based on this comparison, such as minimizing an excess of cloud-ice in the model simulation. The amount of dust east of Qatar over the water appears to be underrepresented in this model forecast.

### 4.2.3  Objective measures

In advanced validation and data assimilation applications (Section 4.3) an objective measure is needed for the comparison of observed and simulated imagery. For simple measures of similarity, cloud masks can be derived from both a SWIm and a corresponding camera image, using for example sky color (e.g. red/blue intensity ratios). Categorical skill scores can then be used to assess the similarity of the angular or horizontal location of the clouds.

To assess the spatial coherence of image values (thus radiances) between the simulated and observed images, the Pearson correlation coefficient $r$ can be determined as

$$r = \frac{N \sum xy - \sum x \sum y}{\sqrt{[N \sum x^2 - (\sum x)^2][N \sum y^2 - (\sum y)^2]}} \quad , \tag{37}$$

where $N$ is the number of pixel pairs and $x, y$ are the pixel pair values. The mean value of $r$, calculated individually for the set of simulated vs. observed pixel intensities in each of the image channels R, G, B, is denoted as $\bar{r}$. We consider this to be a measure of overall image similarity. The R channel is generally most sensitive to clouds and large aerosols, with blue emphasizing Rayleigh scattering contributions from air molecules and Mie scattering from small aerosols. The G channel is sensitive to land surface vegetation and sky colors that can occur around sunset and twilight. Over many cases of SWIm vs. camera image comparisons, $\bar{r}$ was found to correspond well to the subjective assessment of the sky spectral radiance patterns, circumventing potential bias arising due to a lack of radiance calibration in many types of cameras. Note that $\bar{r}$ values are shown for image comparisons presented in Figs. 11 and 14.

in addition to feature characteristics and locations, $\bar{r}$ values are also affected by how realistic the optical and microphysical properties of the analyzed clouds and aerosols are. In other words, when $\bar{r} < 1$, this reflects possible deficiencies in the quality of (i) the 3D digital analysis or specification of hydrometeors, aerosols, and other variables; (ii) the calibration of observed camera images, and (iii) the realism or fidelity of the SWIm algorithms. Recognizing that (a) with all their details, visible imagery is high dimensional and good matches are extremely unlikely to occur by chance, and that (b) high $\bar{r}$ values attest to good performance in all three aspects listed above (i, ii, and iii), the occurrence of just a few cases with high $\bar{r}$, as long as they span various atmospheric, lighting, and observing position conditions, may be sufficient to demonstrate the realism of the SWIm algorithms. For example, the correlation coefficient between the two images in figure 11 is 0.961, indicating the smoke induced aureole around the sun (caused by forward scattering) is well depicted by SWIm. To improve the accuracy of the $\bar{r}$ metric in future investigations we are instituting a $5°$ exclusion radius around the sun to mask out lens flare.

**4.3 Assimilation of camera and satellite imagery**

Today, NWP model forecasts predominate most weather prediction applications from the hourly to the seasonal time scales. Fine scale (up to 1 km) nowcasting in the 0-60 or -120 minutes time range is the notable exception. It cannot even be evaluated whether numerical models lack realism on such fine scales as relevant observations are sporadic and no reliable 3D analyses are available on those scales, which would also be needed for successful predictions. No wonder: NWP forecasts are subpar compared with statistical or subjective methods in hazardous weather warning applications. It is a catch 22 situation: model development is hard without a good analysis, and quality

analysis is challenging to do without a good model - this is the latest frontier of NWP development. The comparisons presented in Figs. 10 and 12 offer a glimmer of hope that model evaluation and initialization may one day be possible with advanced and computationally very efficient tools prototyped in a simple fashion with SWIm and LAPS as examples.

With new geostationary satellite instruments (e.g. ABI) now available, an abundance of high-resolution satellite data are available in spatial, temporal, and spectral domains. As ground-based camera networks also become more readily available we envision a unified assimilation of camera, satellite, radar, and other, more traditional and new data sets in NWP models. SWIm can be used with camera images (and possibly visible satellite images) as a forward operator to constrain model fields in a variational minimization. One approach entails the development and use of SWIm's Jacobian or adjoint, while other techniques employ recursive minimization. Vukicevic et al., 2004 and Polkinghorne and Vukicevic, 2011 proposed to assimilate infrared and visible satellite data using 3D- and 4DVAR methods. Likewise, observed camera images can also be assimilated within a 3/4DVAR cloud analysis module. Such capabilities may be useful in NWP systems such as GSI, the Joint Effort for Data assimilation Integration (JEDI, https://www.jcsda.org/jcsda-project-jedi), vLAPS (Jiang et al., 2015), or other systems.

SWIm can be used in conjunction with other forward operators (such as the CRTM and SHDOM, to compare simulated with observational ground, air, or space based camera data in various wavelengths or applications. Along with additional types of observations (e.g., RADAR, METARs) and model physical, statistical, and dynamical constraints (e.g., using the Jacobian or adjoint), a more complete 3-D and 4-D variational assimilation scheme can be constructed to initialize very fine scale cloud-resolving models. Such initial conditions may be more consistent with full resolution radar and satellite data. Note that on the coarser, synoptic and sub-synoptic scales, adjoint-based 4D variational data assimilation (DA) methods such as that developed at the European Center for Medium Range Forecasts (ECMWF) proved superior to alternative, ensemble-based DA formulations. The authors are not aware of any credible arguments for why this would not also be the case for cloud scale initialization.

A variational 3D tomographic analysis highlighting precipitating hydrometeors was performed with airborne passive microwave observations (Zhou et al., 2014).
In recent years several groups have experimented with extraction and use of cloud information from camera images.An example solving for a 3D cloud mask using a ground-based camera network as discussed in (Viekherman et al., 2014). This has

been expanded using airborne camera image radiances to perform a 3D cloud liquid analysis (Levis, Schechner, Aides, 2015; Levis, Schechner et al., 2015) using a similar forward operator (SHDOM) in a variational solver using a recursive minimization. A corresponding aerosol Observation Simulation Experiment OSE analysis (Aides et al., 2013) was also performed with a ground-based camera network. A design for tomographic camera-based cloud analysis has more recently been developed (Mejia et. al, 2018).

As an initial non-variational test, the authors experimented with the use of the $\bar{r}$ metric described in Section 4.2.3 above. This involves clearing existing, or adding new clouds based on cloud masks derived from color ratios seen in the simulated and/or actual camera images. A single iteration of an algorithm to modify the 3D cloud fields with the mask information often yields improvement in $\bar{r}$ judging from a series of real-time case studies. The removal of clouds just above the reference point, and additions in South and NNW direction resulted in increase of $\bar{r}$ from 0.407 to 0.705 in the example of Fig. 14. This improvement is consistent with visual inspection of clouds between the camera image (b) and the modified simulated image (c) vs. the simulated image from an analysis without the use of the ground-based camera image (a).

Since SWIm operates in three dimensions and considers multiple scattering of visible light photons within clouds it can help perform a 3D tomographic cloud analysis. To move towards the goal of comparing observed and simulated absolute radiance values in a variational setting, two strategies are being considered. The first strategy would entail more precise calibration of camera exposure and contrast so images can be directly compared using a root mean square statistic. A second strategy entails using the simulated image to estimate Global Horizontal Irradiance (GHI, Section 3.1) and then comparing with a GHI measurement made with a pyranometer colocated with the camera.

## 5. Discussion and Conclusion

To make SWIm more generally applicable, its ray tracing algorithms have been extended to address simulations with various light sources, optical phenomena (e.g. rainbows), and twilight colors (to be reported in future publications). Current SWIm development is focused on aerosol optical properties and multiple scattering. Ongoing work also includes refinements to the single scattering albedo and the phase function for various types of aerosols, including dust and smoke. The parameterization being used to determine effective multiple scattering albedo $\omega'$ is being revised to improve

reflectance values associated with thick dust and smoke seen from space-based vantage points. Concurrently the improved parameterization of absorption with multiple-scattering will determine how dark it becomes for ground-based observers when heavy smoke and/or thick dust is present. Under these conditions, spectral variations in $\omega'$ become amplified as $\tau$ increases, causing the sky to have more saturated colors as it darkens.

A fast 3-D radiative transfer model in visible wavelength with a corresponding visualization package called Simulated Weather Imagery (SWIm) has been presented. As summarized in Table 1, SWIm produces radiances in a wide variety of situations involving sky conditions, light sources, and vantage points. Even though other packages are more rigorous for particular situations they are designed for, that comes at a significantly higher computational cost. The visually realistic SWIm color imagery of weather and land surface conditions makes the complex and abstract 3D NWP analyses and forecasts from which it is simulated from perceptually accessible, facilitating both subjective and objective assessment of NWP products. Initial use of SWIm has emphasized its role as a realistic visualization tool. Ongoing development and evaluation will allow SWIm to be used in a more quantitative manner in an increasing variety of situations. To date the evaluation has focused mainly on comparisons with ground-based cameras, pyranometers, and DSCOVR imagery, even though they typically include the LAPS cloud analysis used for SWIm input in the evaluation pipeline. Specific comparisons with other radiative transfer packages (e.g. CRTM, MYSTIC) is a good topic for future work.

Validation of SWIm is summarized in Table 4 and consists of both qualitative and quantitative assessment. The quality of the hydrometeor and aerosol analysis plays a role, making these joint comparisons of SWIm and the analysis techniques. Additional quantitative validation is planned to compare SWIm with other 1D and 3D radiative transfer models in a manner that is more independent of analysis quality.

Simulated time-lapse sky camera views for both recent and future weather can be used, for example, for the interpretation and communication of weather information to the public  (an archive of near real-time examples available at http://stevealbers.net/allsky/allsky.html) Interactive 3D flythroughs viewed from both inside and above the model domain can be another exciting way to display NWP model results for both scientific and lay audiences. This includes the use of in flight simulators for aviation purposes, along with other interactive game engines. High quality images or animations from existing or to be installed all-sky cameras with greater than 180° field of view at official meteorological or other observation sites could also be used to evaluate

clouds, aerosols, and land surface features such as snow cover analyzed or forecast in NWP systems.

A critical use of camera images in the future will be their variational assimilation into high-resolution analysis states for the initialization of NWP forecasts used in Warn-On-Forecasting (Stensrud et al., 2013). The comparison of high quality ground-, air-, or space-based camera imagery with their simulated counterparts  is a critical first step in the assimilation of such observations. The assimilation of such gap-filling observations can be especially useful in pre-convective environments where cumulus clouds are present while radar echoes have yet to develop. Today's DA techniques suffer in such situations, severely limiting the predictability of tornadoes and other high impact events. 4-D variational tomographic DA is designed to combine camera and satellite imagery from multiple viewpoints. The sensitive dependence of multiple scattering in 3D visible wavelength light propagation on the type and distribution of hydrometeors facilitates a better initialization of cloud properties throughout the depth of the clouds. This in turn can potentially extend the time span of predictability for severe weather events from the current period starting with the emergence of organized radar echoes back to the more subtle beginnings of cloud formation.

 As the spatiotemporal and spectral resolution of color imagery observed both with ground-based cameras or air- and satellite-borne instruments and corresponding output from NWP models reaches unprecedented highs, a question arises whether variational or other DA methods can sensibly combine information from the two sources? If they can, consistent analyses of clouds and related precipitation and aerosol fields will aid situational awareness and fine-scale model initialization. SWIm used as a 3-D forward operator for camera and visible satellite imagery may help addressing the above and related challenges.

**Acknowledgements:**

This work was partially funded by a Multidisciplinary University Research Initiative (MURI) called Holistic Analysis of Aerosols in Littoral Environments (HAALE). For the HAALE-MURI project the support of the Office of Naval Research under grant N00014-16-1-2040 is gratefully acknowledged. Additional funding was provided by NOAA under the Cooperative Institute for Research in the Atmosphere (CIRA). We thank Didier Tanre and the AERONET team for establishing and maintaining Capo Verde AERONET site used in this investigation. We also thank Afshin Andreas and Mark Kutchenreider of the National Renewable Energy Laboratory (NREL) in Golden Colorado, along with Will Beuttell of EKO Instruments Inc. for help in accessing their

real-time all-sky camera images. We appreciate the helpful feedback and suggestions provided by two anonymous reviewers.

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

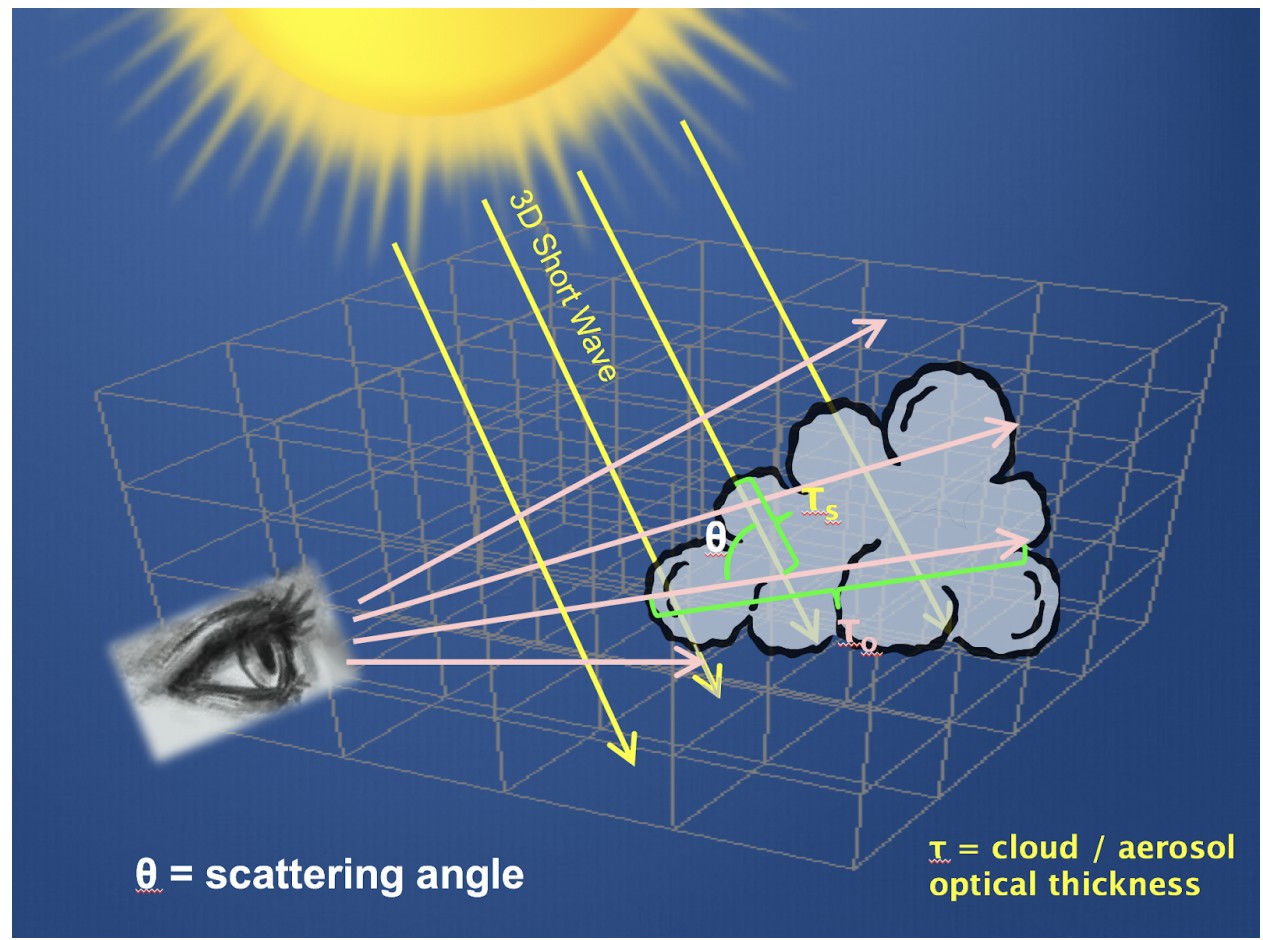

Figure 1. General ray-tracing procedure showing forward light rays (yellow) coming from the light source. A second set of light rays (pink) are traced backward from the observer. The forward and backward optical thicknesses ($\tau_s$ and $\tau_o$) are calculated along these lines of sight and used for subsequent calculations to estimate the radiance on an angular grid as seen by the observer.

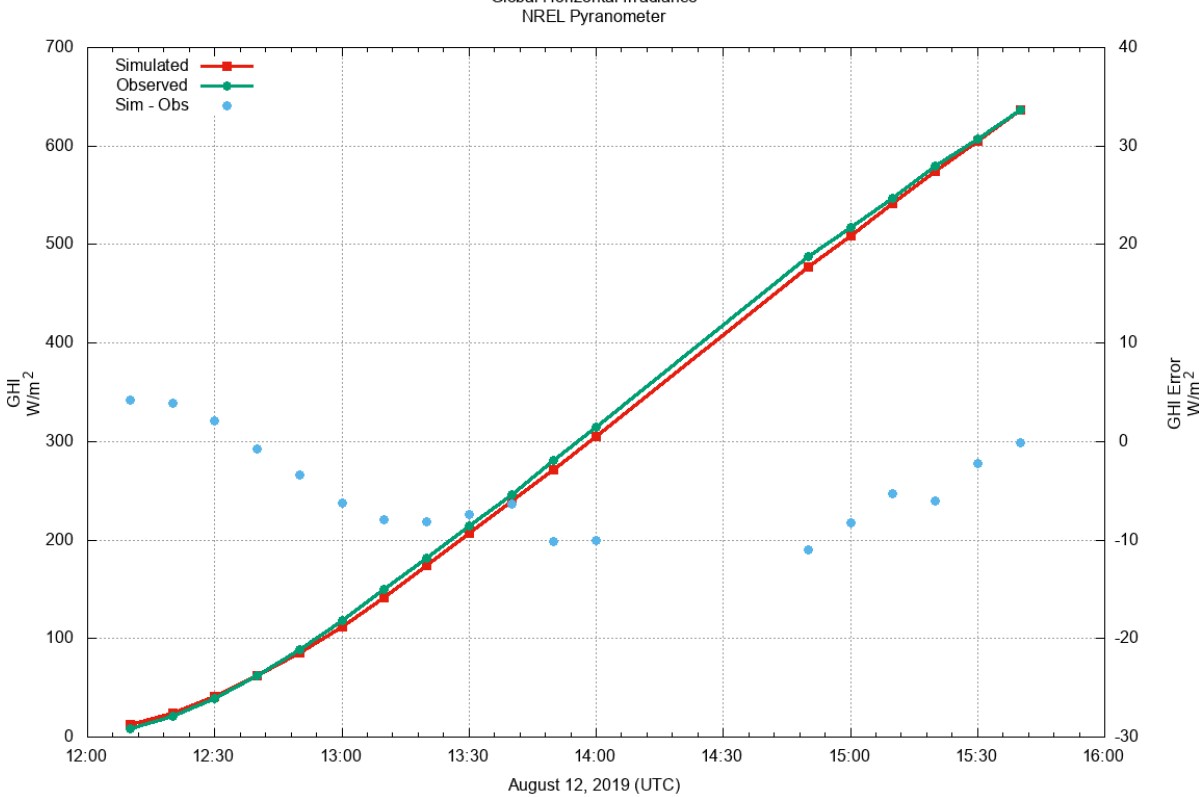

Figure 2. Time series of GHI values integrated from SWIm radiance images (red lines, vertical axis on left) compared with concurrent pyranometer observations in $Wm^{-2}$ at NREL (green lines). The comparison spans a 4 hour period on the morning of August 12, 2019. Simulated minus pyranometer GHI values are plotted as blue circles (vertical axis on right). Sky conditions were free of significant clouds, with aerosol optical depth < 0.1.

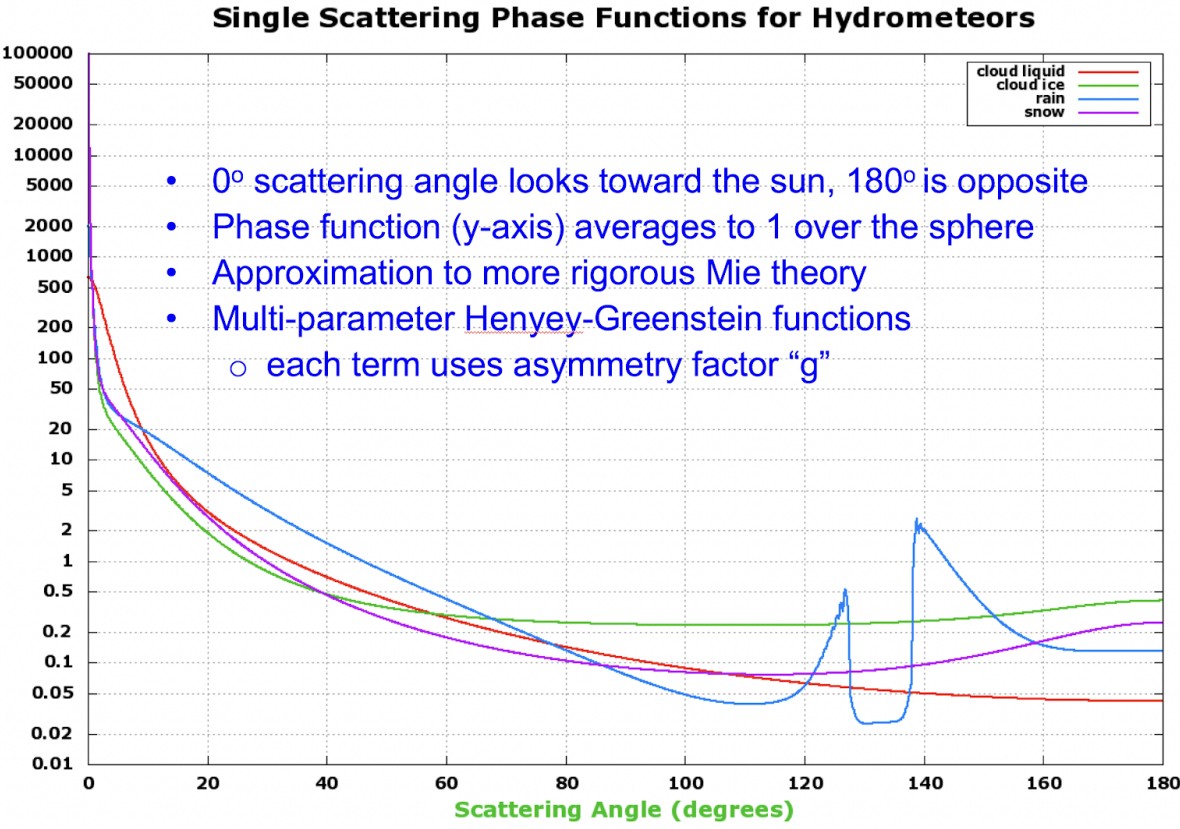

Figure 3. Single scattering phase functions used for cloud liquid, cloud ice, rain, and snow.

# Aerosol Phase Functions

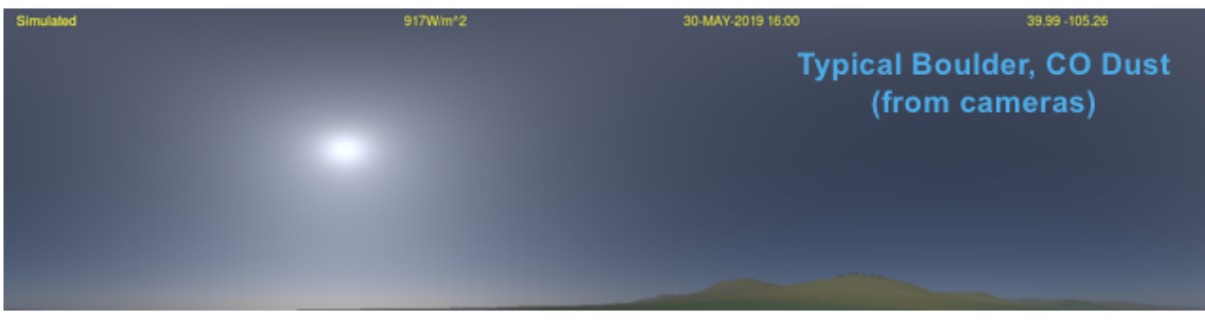

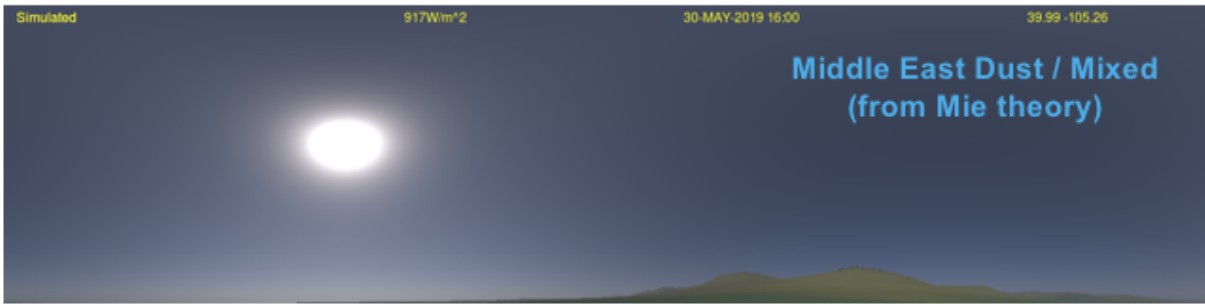

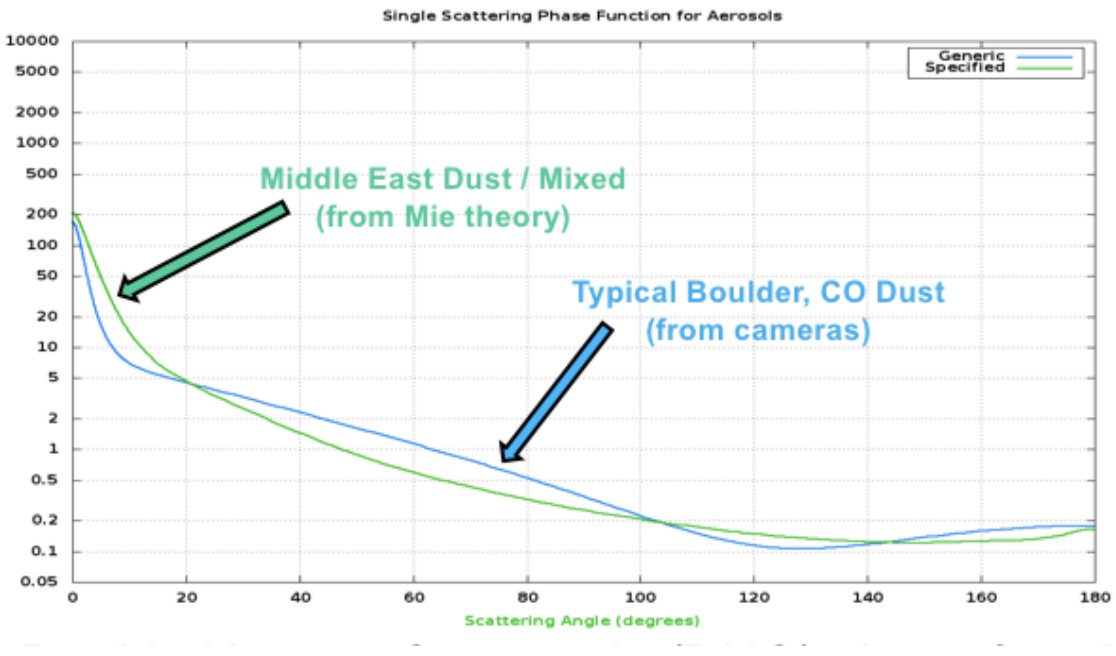

Figure 4. Simulated panoramic images with an AOD of 0.1 using the Colorado empirical phase function (a), and the Mie theory mixed dust case (b). These two phase functions are compared in (c).

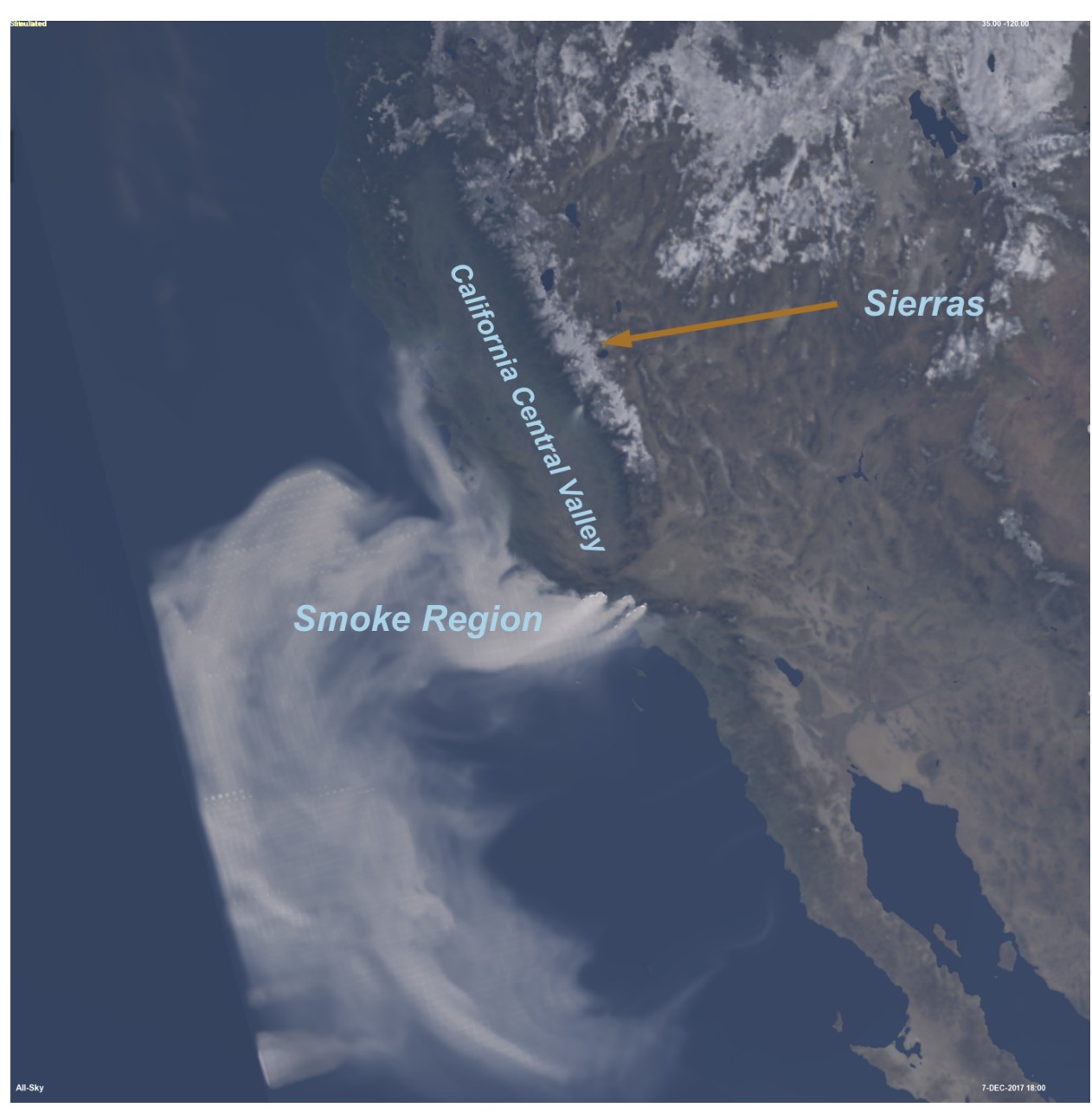

Figure 5. Simulated image of a HRRR-Smoke forecast with a smoke plume from the December 2017 California wildfires. The view is zoomed in from a perspective point at $40000\ km$ altitude.

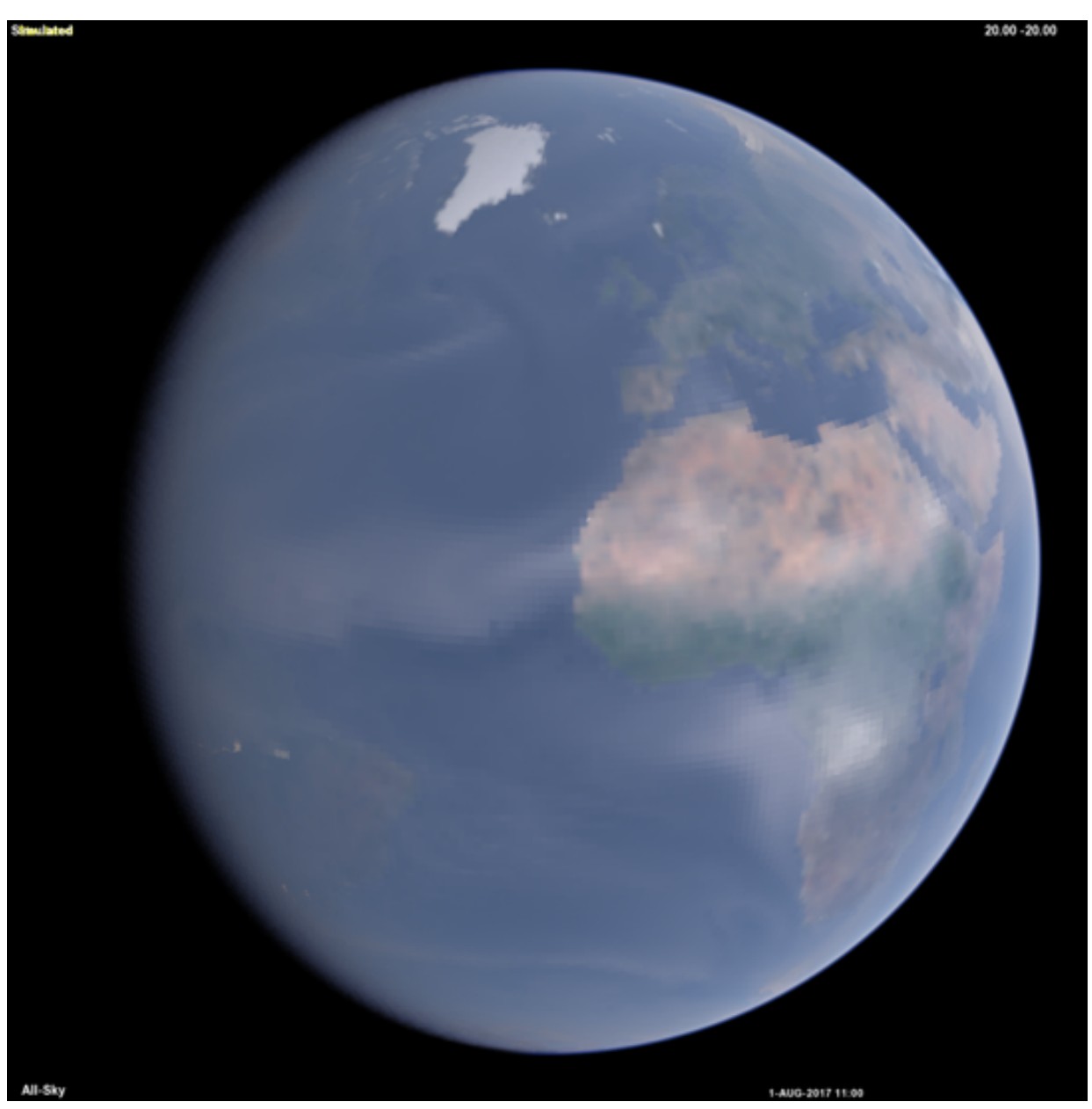

Figure 6. View from space of the NAVGEM global model, using aerosols only. The perspective point is $1.5 \times 10^6 km$ distant.

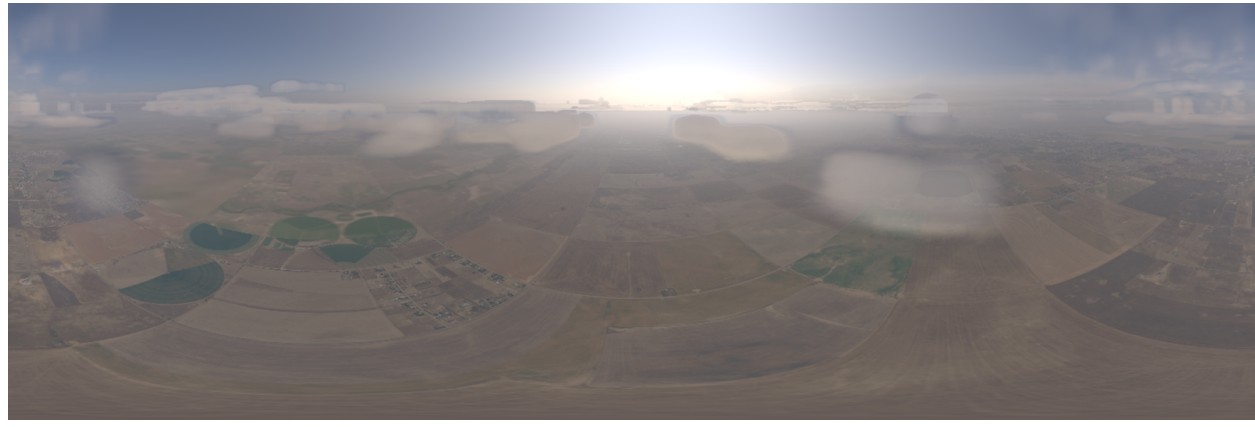

Figure 7. In-situ panoramic view in the lower troposphere showing smoke aerosols and hydrometeors. This is part of an animation simulating an airplane landing at the Denver International Airport. The panorama spans $360^o$ from a perspective $\sim 4km$ above ground. Hydrometeor fields are from a LAPS analysis.

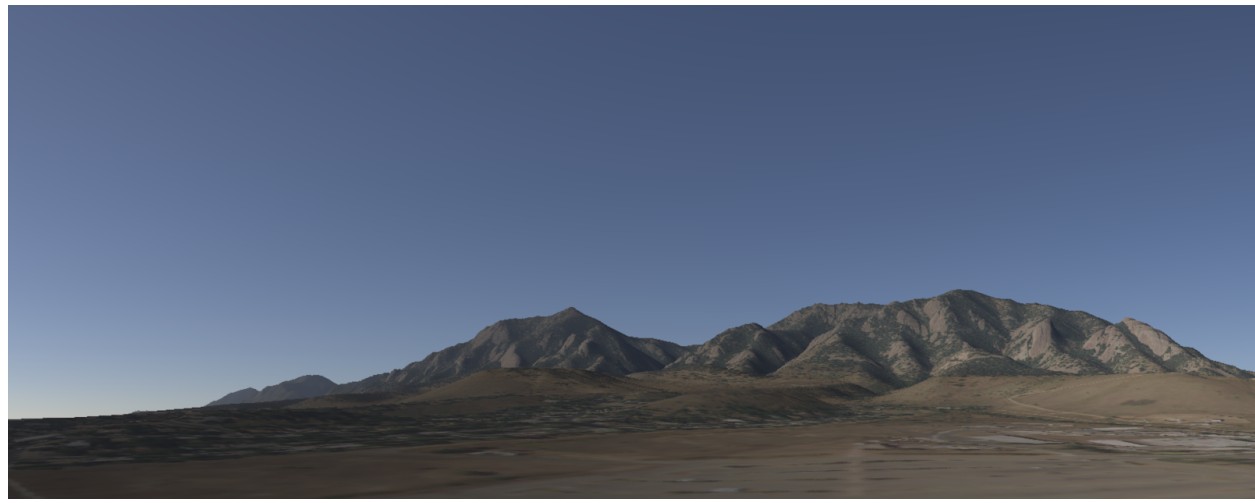

Figure 8. SWIm generated image for a hypothetical clear-sky case having an aerosol optical depth ~0.05. The model grid and associated terrain data is at 30m resolution and surface spectral albedo information is derived from 0.7m resolution aerial imagery from the USDA. The vantage point is from the U.S. Department of Commerce campus in Boulder, Colorado, looking at azimuths from south through west.

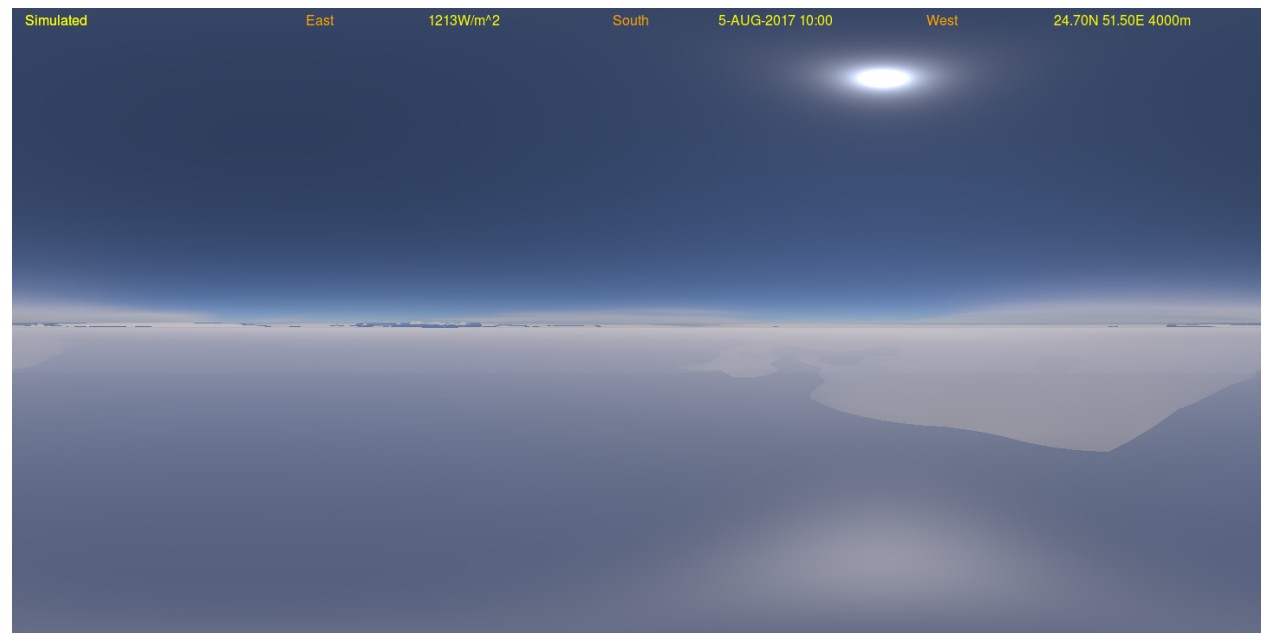

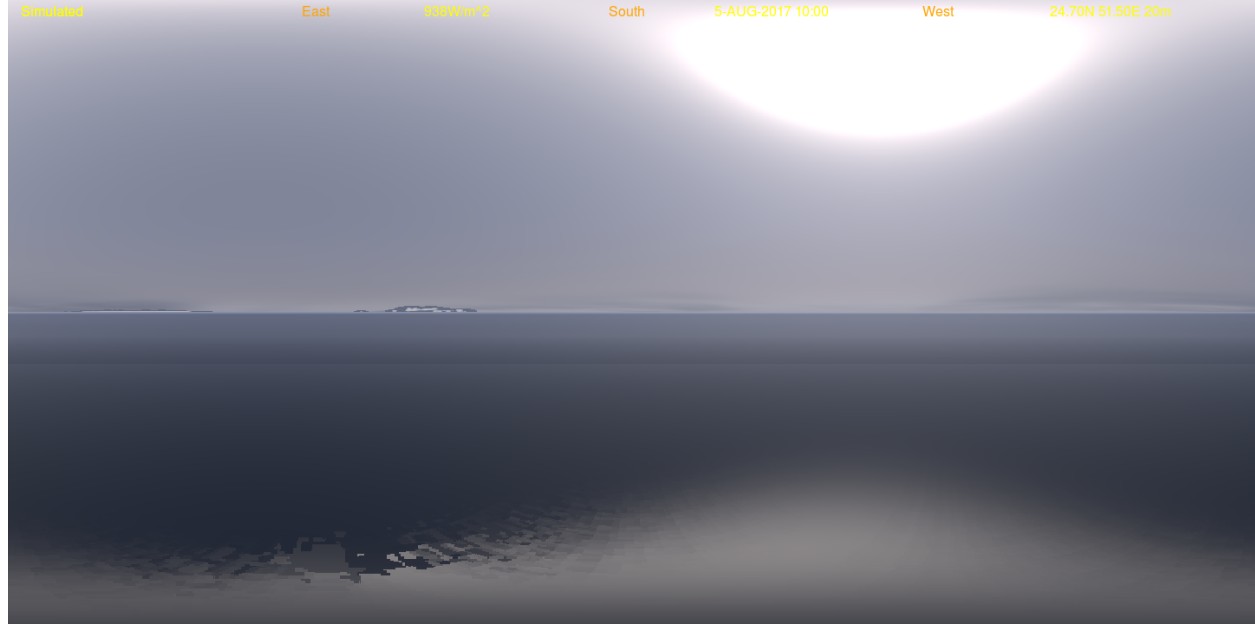

Figure 9. View from $4km$ $(a)$ $and$ $20m$ (b) above the Persian Gulf of a RAMS model simulation showing dust, hydrometeors, land surface, and water including sun glint, displayed with a cylindrical (panoramic) projection.

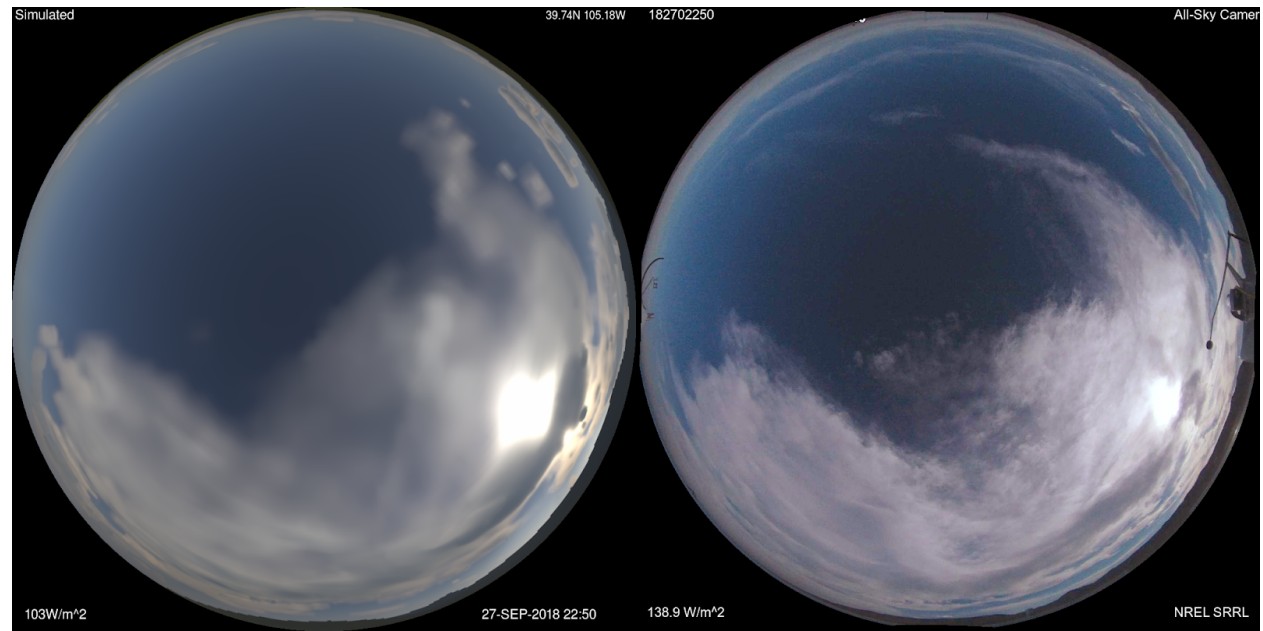

Figure 10. Comparison of observed (right) to simulated (left) polar equidistant projection images showing the upward looking hemisphere from a ground-based location in Golden, Colorado on September 27, 2018 at 2250UTC. LAPS analysis fields are used for the simulated images.

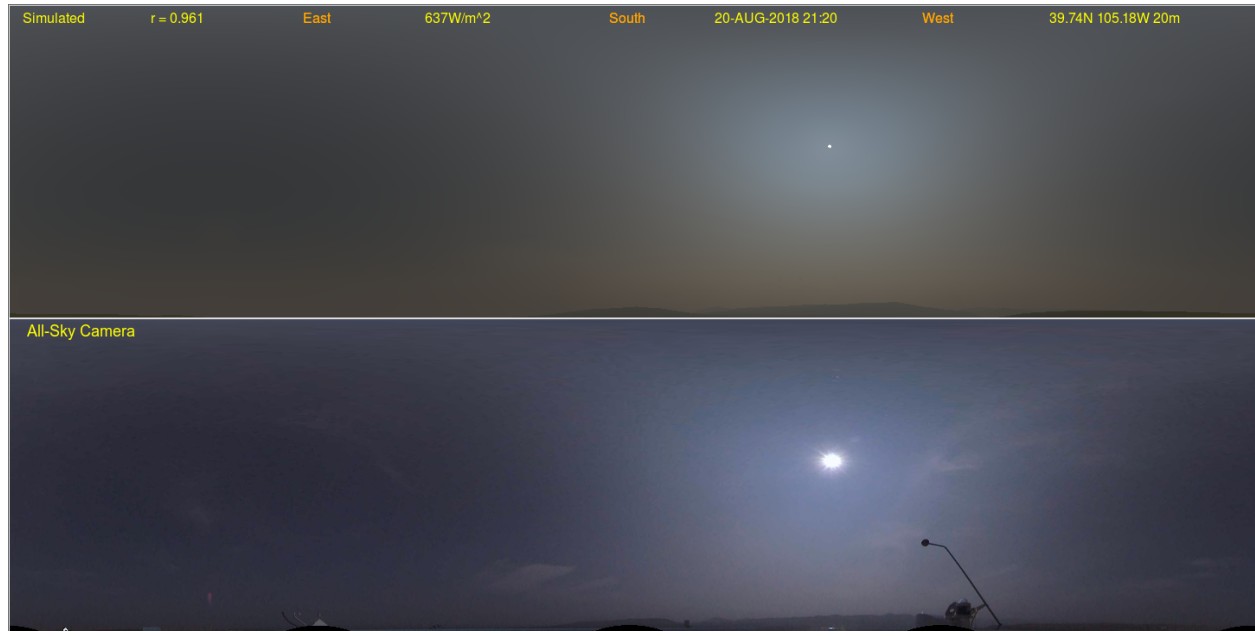

Figure 11. A comparison of aerosols at 2100UTC on August 20, 2018 in Golden, Colorado showing a panoramic simulated (top) and an all-sky camera image (bottom). The correlation $\bar{r}$ between the images is denoted as 0.961.

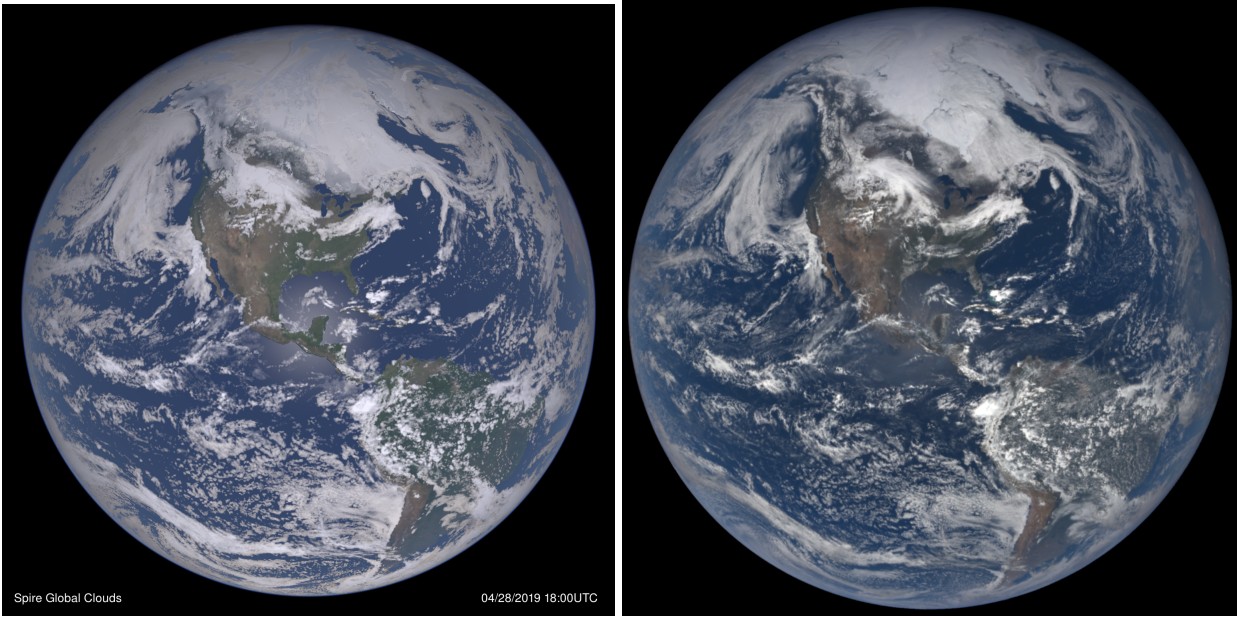

Figure 12. Side-by-side comparison of global cloud coverage viewed from space at approximately 1800UTC on April 28, 2019 as provided by DSCOVR-EPIC (camera observed image, right), and analyzed by LAPS (21 km horizontal resolution) and visualized by SWIm (simulated image, left).

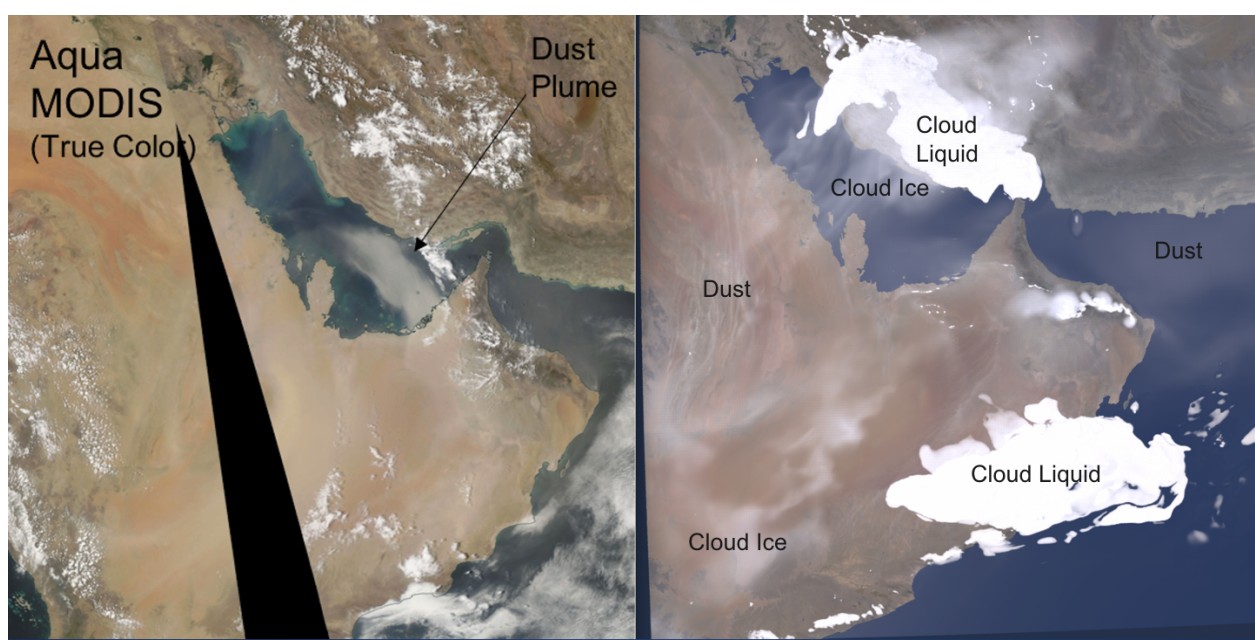

Figure 13. Aqua-MODIS image (left) taken from passes at about 1330 local time over the Arabian Peninsula compared with SWIm visualization of a RAMS model forecast (right) from 1000UTC. Areas having predominantly dust, cloud liquid, and cloud ice are annotated in the images.

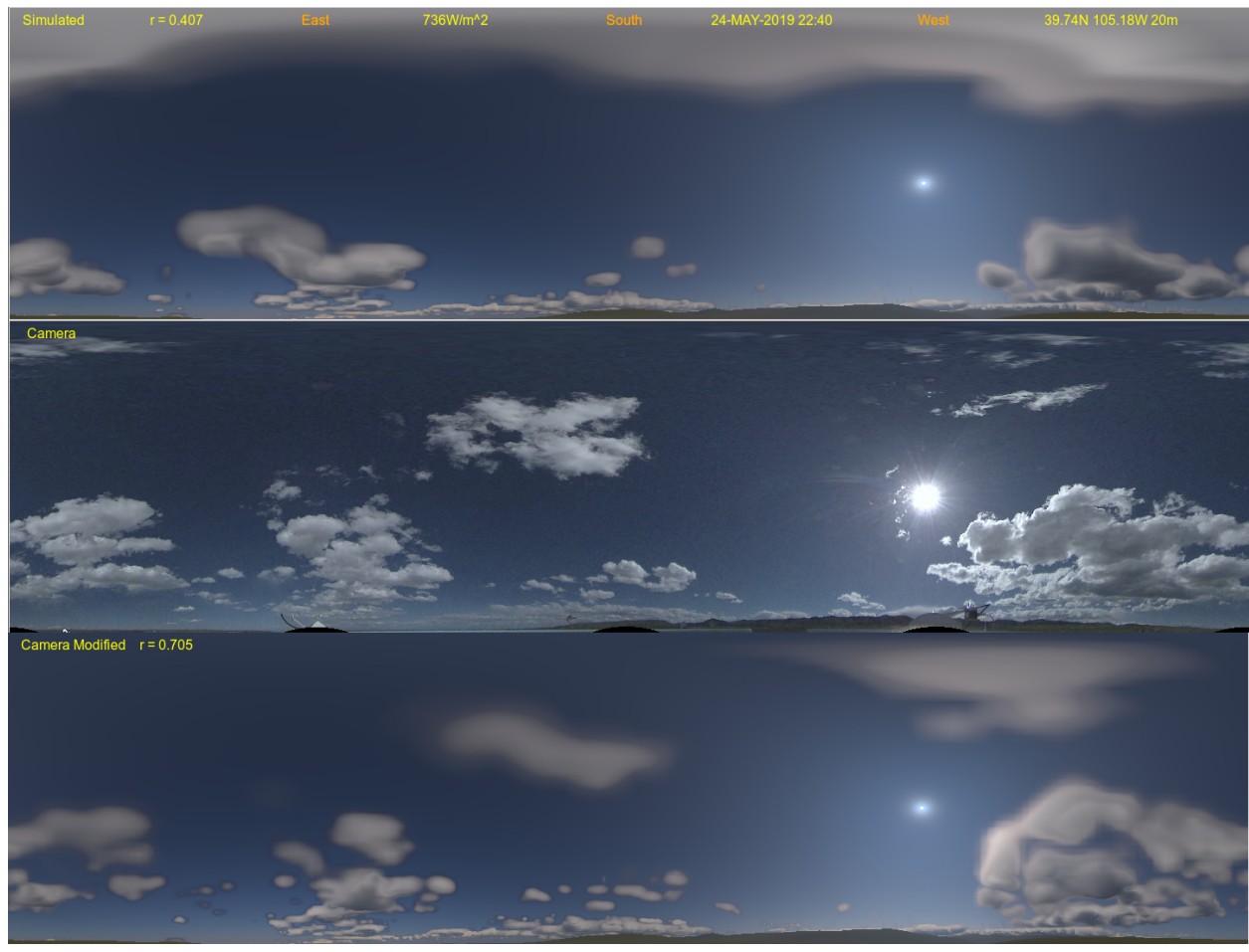

Figure 14. SWim image from a 3D LAPS cloud analysis using satellite data without camera input (a), is shown with a camera image (b), and the SWIm image using 3D clouds modified via a color ratio algorithm (c). The NREL camera image is from May 24, 2019 at 2240UTC.

| | SWIm | CRTM | RRTMG | SHDOM | Monte Carlo |
|---|---|---|---|---|---|
| 3-D Radiation (including sideways) between columns | Yes | No | No | Yes | Yes |
| Multiple Scattering | Approximate | Yes | Yes | Yes | Yes |
| Fast Running | Yes | Yes | Yes | No | No |
| Ground- air- or space-based observer | All | Space | Space | All | All |
| Curved Earth Shadow / Twilight | Yes | No | No | | Yes |
| Moon / Stars / City Lights | Yes | No | No | | |
| 2-D (Directional) Images | Yes | Yes | TOA SW up (Isotropic) | Yes | Yes |
| Wavelengths | Visible | Vis + IR | Vis + IR | | |
| Grid Resolutions | All | All | All | $\leq 100m$ | All |

Table 1. Overview of functionality in a sampling of radiative transfer packages.

| Step |
|------|
| Step 1a: Forward rays from dominant light source (in 3-D grid, including hydrometeors and aerosols) |
| Step 1b: Backward rays from observer (in 3-D grid, including hydrometeors and aerosols) |
| Step 2: Rays from Sun and from observer (in clear air, extending beyond model grid) |
| Step 3: Combination of radiance components, generation of RGB image display. |

Table 2. List of ray tracing steps used in SWIm. Steps 1a and 1b are illustrated in Fig. 1.

| Case | $g_1$ | $g_2$ | $f_c$ | $f_b$ | $\omega$ |
|------|-------|-------|-------|-------|----------|
| Colorado Dust | .59, .60, .61 | .895, .900, .905 | .12, .12, .12 | .550, .550, .550 | .935, .92, .86 |
| Saudi Arabian Mixed Dust and Pollution | .23, .27, .29 | .915, .925, .933 | .58, .54, .53 | .562, .558, .558 | .96, .96, .96 |

Table 3. Two cases showing the four fitted phase function parameters $g_1$ , $g_2$ , $f_c$ , and $f_b$ as well as single scattering albedo $\omega$ , for each of the three reference wavelengths, 615nm, 546nm and 450nm.

| Quantity being assessed | Measurements | Methodology | Outcome / Result | Comments |
|---|---|---|---|---|
| GHI | NREL pyranometer | 546nm horizontal spectral radiance integrated over sky dome, converted to global horizontal irradiance. | Typically within $10-20 \frac{W}{m^2}$ in cloud-free skies. SWIm ~50% too high in uniform overcast. | Sensitive to both SWIm raytracing, and cloud/aerosol analysis. |
| Spatially (radially) distributed spectral radiance (converted to RGB images) from surface vantage point | NREL all-sky camera | Correlation ($\bar{r}$) (described in text) calculated over sky dome between concurrent SWIm and camera RGB images. | Typically 0.90 to 0.98 in cloud-free areas (where aerosols remain important) and ~0.50 with significant cloud cover. | Higher scores contingent on masking 12 degree radius around sun affected by camera glare.<br><br>Cloudy results strongly affected by quality of cloud (and to lesser degree, aerosol) analysis, and thus highly variable; in best cases, correlation reaches ~0.8. |
| Spatially distributed images from space | DSCOVR EPIC RGB images and red band reflectance factor data | Subjective comparison of SWIm and concurrent DSCOVR/EPIC data | Reflectance factor distribution matches anticipated values from 5% in darkest clear oceanic areas to ~1.1 in bright tropical convection. | Results sensitive to analysis quality of clouds (and aerosols), whose locations are well captured both on large and small scales. |

Table 4. List of SWIm validation methods being developed.

**Appendix A. Aerosol optical properties for Arabian peninsula case.**

The Arabian Peninsula case is calculated using the representative dust model derived as follows from the Capo Verde site in the AERONET network (Holben et al., 1998). We the applied EPA positive matrix factorization (PMF) 5.0 model (available at https://www.epa.gov/air-research/positive-matrix-factorization-model-environmental-data-analyses) to the dataset, using as factors the aerosol optical depth (AOD) for the fine and coarse modes and the total absorption aerosol optical depth (AAOD) from the Capo Verde site, for all Level 2.0 Inversion V3 data from 1994-2017. Two factors were derived (Figure A1). The factor with high AOD contributions from the coarse mode was flagged as the dust source. The derived absorption angstrom exponent (AAE) for Factor 1 was 4.387 for the Capo Verde site and the average extinction Ångstrom exponent (EAE) was 0.0905, lying in the range of the dust aerosol characteristics identified in Giles et al. (2012). The factor with high AAOD was believed to be associated with urban / industrial aerosols. For those samples, the averaged AAE and EAE were 0.729 and 1.164, respectively, similar to reported optical properties of absorbing fine particles (Giles et al., 2012). We selected data with corresponding PMF-identified dust source contributions larger than 95% to characterize the dust properties. The average normalized volume size distributions for the dusty days is shown in Figure A2. We used the average retrieved refractive index for the same dusty days, and the aspect ratio distribution in Dubovik et al (2006), to calculate the phase function and related optical properties used in this study.

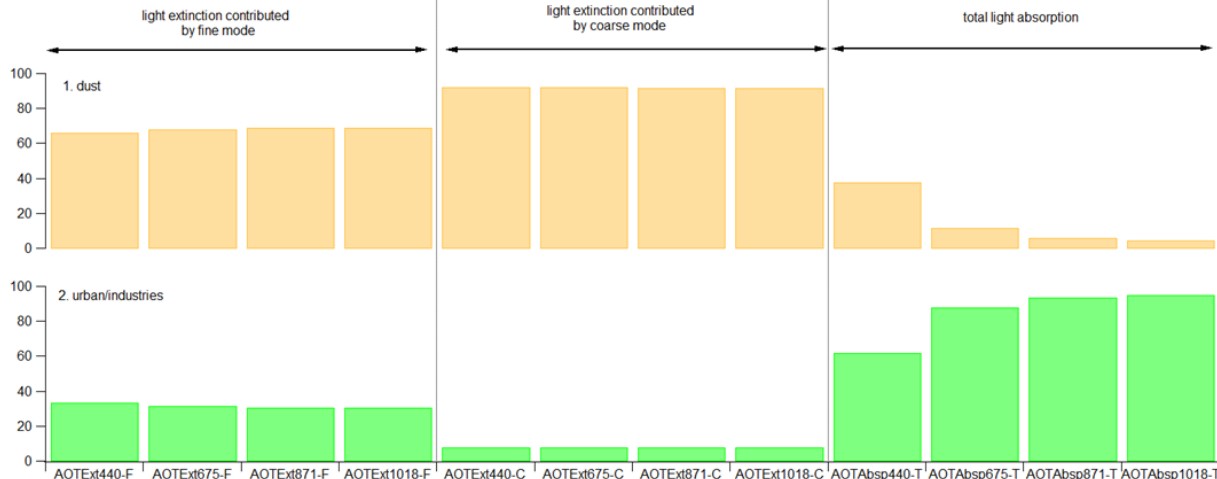

Figure A1. Optical source profile (% of species in each source) for the Capo Verde dataset.

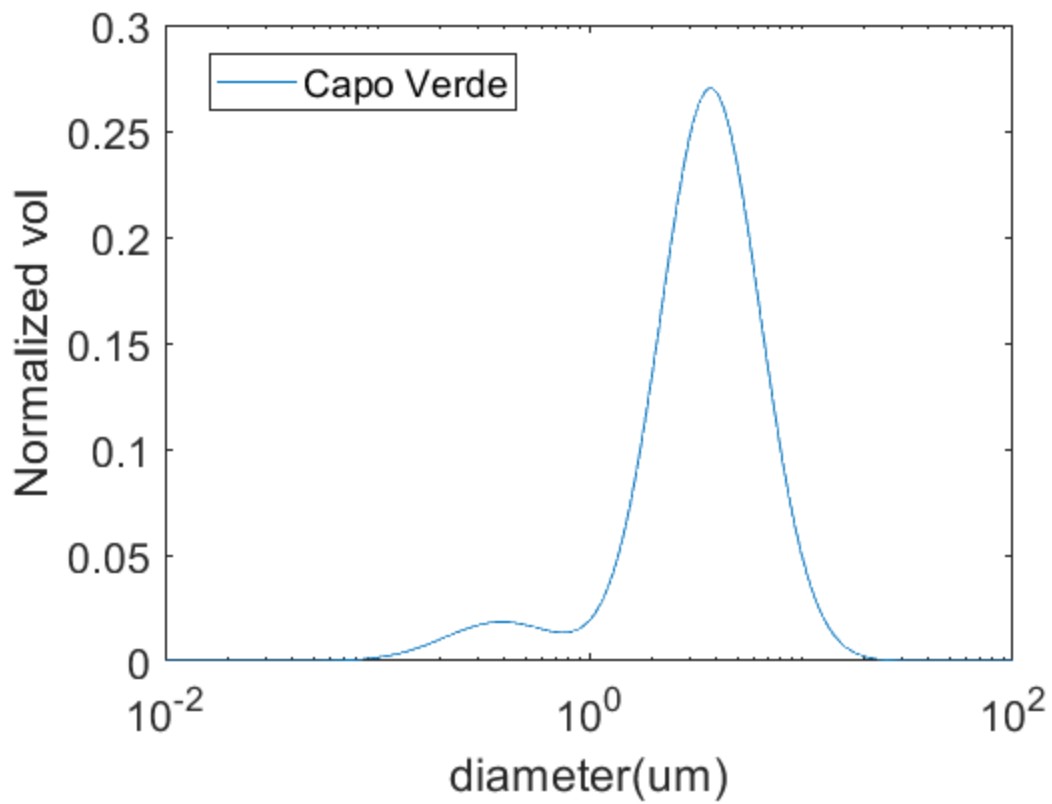

Figure A2. Average normalized volume size distribution for dust-dominated days in the Capo Verde data set.

**Appendix B. Multiple scattering effective phase functions for additional species.**

For multiple scattering for hydrometeors beyond cloud liquid we follow a procedure
similar to that described in section 3.4.2 with these primary differences. For the rain
phase function we specify via eq. 13 a parameterization for multiple scattering. The
optically thin rain component is given here:

$$P_{thin}(\theta,\lambda) = 0.1\, p(\theta,0.99^{\tau_o}) \;+\; 1.05\, p(\theta,0.75^{\tau_o}) \;-\; 0.35\, p(\theta,0.0) \;+ 0.20\, p(\theta,-0.2) \quad \text{(B1)}$$

If there is a mixture of cloud liquid and rain then we interpolate between the results of
eqs. 18 and B1.

For cloud ice, the optically thin component is given by.

$$P_{thin}(\theta,\lambda) = 0.50\, p(\theta,0.999^{\tau_o}) \;+\; 0.71\, p(\theta,0.991^{\tau_o}) \;-\; 0.25\, p(\theta,0.0) \;+ 0.04\, p(\theta,-0.2) \quad \text{(B2)}$$

For snow eq. B3 is used. If there is a mixture of cloud ice and snow then we interpolate
between the results of eqs. B2 and B3.

$$P_{thin}(\theta,\lambda) = 0.50\, p(\theta,0.999^{\tau_o}) \;+\; 0.45\, p(\theta,0.991^{\tau_o}) \;+\; 0.03\, p(\theta,0.0) \;+ 0.02\, p(\theta,-0.2) \quad \text{(B3)}$$