# Peer review of "A Fast Visible Wavelength 3-D Radiative Transfer Model for Numerical Weather Prediction Visualization and Forward Modeling"

_Atmospheric Measurement Techniques, 2019_

## Referee Comment (RC1) · Anonymous Referee #1 · 20 Jun 2019

The manuscript presents a fast, approximate 3D radiative transfer procedure for visualization of numerical weather prediction and forward modeling of ground-, aircraft-, and satellite-based camera and imager. Reflection by optically thick cloud or aerosol layer is parameterized. The scattering phase function of cloud and aerosol is approximated by the double Henyey-Greenstein function. The model is capable of produce true-color images composed of radiances at three wavelengths in the visible spectrum. While the authors emphasized the usefulness of the model in data assimilation in weather models, future works are needed to actually apply to data assimilation. Technical details of the calculation method should be clarified. I believe that important improvements are needed in the manuscript. My recommendation is therefore to make major revisions.

Please find below my detailed comments.

General comments

1. A lot of technical details are missing. For example, a) authors mentioned about a combination of single and multiple scattering components of radiance and some interpolation method between optically thin and thick regimes, but the methods for the combination and interpolation are not given. b) Actually, what equations are used to calculate the total radiance? It is unclear how radiances are calculated when interactions between clouds and surface reflection is present. Please show equations of total radiance that accumulates all contributions from the atmosphere, clouds, aerosols, and surface. c) The method for determining "a single scattering phase function that is equivalent to the net effect of the multiple scattering events" is missing in the current manuscript. With unknown methods, I cannot judge the validity and values of the "approximations". I have raised more points in the "specific points" below.

2. This is probably because the technical details are obscured, I cannot evaluate the model's accuracy, validity and limitations. How accurate is this model? What approximations are used? These should be clearly stated in Abstract and Conclusions.

3. As indicated in Fig. 2, the model possibly uses the bidirectional (dual-path) ray tracing along lines of sight and paths from radiation source (i.e. typically the sun) to scattering points. However, actually, it is not clear how to compute the radiance. There is no description on evaluation of the integral of scattering contributions from segments of the line of sight. Or, maybe, the model does not actually compute the integral, but instead scattering contributions from aerosol and cloud elements are approximated to be from some single point within cloud or aerosol layer. Even if so, that "point" is unknown.

4. The adjoint of this radiative transfer model is not available, and more works are needed for data assimilation. I understand the adjoint is not necessary in some frameworks. However, this paper's focus is not data assimilation, anyway. Subsection 4.3

just suggests possibilities and outlook by lengthy descriptions lacking any evidence and results. I recommend to greatly shorten this part and merge into the last paragraph of Section 5.

5. Lens flare: Lens flare effects appear in actual total-sky images, which should be discussed if comparing the model simulations with camera images. The lens flare is significant in forward scattering directions and modify the appearance of solar aureole by aerosol particles. This is critical in determining aerosol properties from total-sky images.

6. Subsection 3.2 describes the optional characteristics of the model, which is interesting. However, it is not clear whether the authors are first presenting new modeling of this complicated modeling such as moonlight, city lights, and spherical atmosphere. Are there any previous papers regarding to things presented in this section? If so, the details should be found in those papers, and this paper should just cite the references. In the current form, the descriptions are too short to fully describe the complicated radiative sources such as city light, while no results of demonstration and verification are presented in this paper. If this section should be presented in this paper, supporting information including some examples of nighttime scene and Belt of Venus as the authors mentioned in the text. Anyway, this paper does not focus on such nighttime and twilight cases. I recommend to remove this subsection and to leave them for some separate papers. I hope to see example result, validation, and applications in the future.

Specific points

Title: Does the use of acronym of "NWP" conform to the regulation of AMT?

P1, L16: In my understanding, the proposed procedure is not intended for a radiation scheme in "weather and climate" models. The first paragraph of abstract loses focus.

P5, L20, "This is a unique feature that allows…": The methodology is standard in the

field of atmospheric science, optics and computer vision. For example, MYSTIC, as used in Klinger et al. (2017), uses that standard method. The authors should give appropriate references.

P6: Eqs. (1) and (2) are identical, and there is no reason to present both.

P6, L6, Fig. 2 and Table 1: A method for radiance integration is not clear. In my understanding, the radiance should be an integral of contributions coming from many small segment of line of sight (from camera). Each contribution is a function of irradiance from radiative source, which should be calculated by the forward path using some spatial interpolation or directly tracing a ray from the scattering point to the radiative source. This kind of integration should be explained in detail. Without such a description, the readers cannot understand the relationship between forward and backward rays.

P6, L22, "615nm…": Please give a reason for selection of these 3 wavelengths. Is there any references?

P7, L1, "The light…": The same sentence is given previously.

P7: Eqs. (3) and (4) are identical, and there is no reason to present both.

P9, L4, "A two-stream approach is used to incorporate the backscatter fraction…": This sentence is not clear. There are several two-stream approaches. References should be given. Or, is this paper presenting a new two-stream approach for rendering? The definition of backscatter fraction should be clarified. How is the backward fraction used to determine the total downward illumination? What is "illumination"? Is it equal to the irradiance?

P9, L24: Why are HG functions used? Why not Mie theory? The reason should be explained in the text.

P10, Eq. (6): Explanations of $i$ and theta are missing.

P10, Eq. (7): An explanation of $c_i$ is missing.

P10, L4: "When $\tau \ll 1$": Please give an exact definition of tau. In Eq. (2), tau is optical depth integrated along the beam path. The definition of "beam path" is not clear because there are two paths: from camera to target and from the sun to target. In Eq. (8), how is tau defined? Is it the optical thickness along line of sight, along the direct source beam, or some combination of them? For volumetric object (i.e. cloud), there are infinite number of forward paths (from sun) corresponding to small path segments along the backward path (from camera).

P10, L15-17, "more efficient approximation that arrives at a single scattering phase function...": This approximated phase function is not given in Subsubsection 3.4.2. I also failed to find any description in the manuscript. I guess the phase function should be modified to represent multiple scattering effects. How are they modified from the original single-scattering phase function?

P10, L17-18: The interpolation scheme is not actually given in Subsubsection 3.4.2. I also failed to find any details of this interpolation in the manuscript. What interpolation scheme is actually used between optically thin and thick clouds? Please show it using equations if possible.

P10, L26: Does "heavy overcast sky" mean sky with 100% cloud cover? How is the radiance parameterized as a function of cloud cover? How is the radiance computed when the sky is not overcast?

P11, L7, "Intermediate values of tau_0 are given empirical phase functions": How are tau_0 and the empirical phase functions related?

P11, L28, "A simple bidirectional...": How is the BRDF developed? Is it based on any measurements or theoretical calculations using rigorous radiative transfer models? It is better to show equations of the ARF.

P12, L9: Actually, what and how is interpolated? Is the cloud albedo (or BRDF) interpolated? Is it linear interpolation with respect to tau? It is better to explain the method

with equations and references (if present).

P13, L25: This paragraph mentioned about single-scattering albedo (for single scattering radiance), but previous explanations say that scattering contributions by aerosols do not depend on the single-scattering albedo (Eq. (8)). In P10, L7, single-scattering radiance is insensitive to the single-scattering albedo: "This relationship applies to hydrometeors as well as aerosols". If so, why single-scattering albedo is discussed in P13.

P14, L6, "The first row in Table 2 was derived semi-empirically for relatively dusty days": How was the camera radiometrically calibrated? Were all pixels of camera image used? Was the circumsolar region excluded or obscured by shadow band or anything? Of course, saturated pixels should be excluded. Are there any references?

P14, L11, "AERONET": References are needed.

P15, L3-7, "As with cloud multiple scattering, a rigorous approach such as Monte-Carlo would consider each scattering event explicitly, though this would be computationally inefficient. ... multiple scattering events.": These 2 sentences are almost identical as previously shown in the hydrometeor subsection. Please make a point clear.

P15, L10 and L21, "eq. 6": The equation number seems wrong.

P15, L11: What are CIRA, RAMS, and WRF? They are explained at page 18, but it is too late. Also, references are needed.

P16, L6, "these quantities are merged together to provide the combined radiance": How are they merged? Is the combination simply a sum of radiance components by clear sky and aerosol/cloud? How does the model implement the transmissivities between the sun and cloud/aerosol and between the cloud/aerosol and the camera? Please show equations if possible.

P17, L2-5: A lot of details and references are missing. What is "relatively simple analytical function"? How is the f "modified"? What a sun glint model is chosen from many previously proposed models? How is "scattering from below the water surface" modeled? If there are references, please cite them. If there is no reference, described the details.

P18, L3, "3x3 transfer matrix": Please provide this matrix explicitly or show a reference. This is because there are several variants of RGB color space.

P19, L2, "A more complete. . .": This sentence is not clear. Is this confirmed by tests or just a guess by the authors? Is there any evidence?

P18, L31, "we've" → "we have"

P18, L31: "Simulated Weather Imagery (SWIm)" → "SWIm" (This is already written at P4, L7 and should not be repeated)

P20, L23, "Local Analysis and Prediction System Â∎ LAPS, Toth et al., 2014)" This should be presented earlier because LAPS appears at P19, L9.

P20, L25, "METAR": What is METAR?

P21, L8, "High Resolution Rapid Refresh (HRRR)" → "HRRR". HRRR first appeared earlier. It should be explained when presented first.

P21, Subsubsection 4.2.1: In this comparison of simulated observed imagery, strong direct sun beam can cause lens flare, which should be taken into account. The lens flare may be significant particularly when strong direct beam is incident to the camera as in Fig. 11. The camera image cannot be directly compared with the simulated image.

P21, L26, "The brightness scaling...": This should be primarily influenced by the lens flare.

P22, L7, "GFS": What is GFS?

P22, L17, "uncertainty in the brightness scaling of the DSCOVR imagery" : Why is it uncertain? If the radiance data are from the EPIC product, the brightness scaling should be the same as for simulated imagery.

P23, L32, "One approach would entail developing SWIM's Jacobian or adjoint, while other techniques employ recursive minimization.": This sentence is not clear. Please give references. The authors stated that "SWIm can be used as a forward operator in a variational minimization". However, the derivatives are required in the variational minimization, while derivative calculations are not in scope of the current manuscript. This is confusing. It is not very clear whether he current SWIM be used for data assimilations or further works are needed.

P23, Subsection 4.3: This section seems to be the authors' outlook. The problem is that it is not clear whether it is outlook or not. It is better to shorten this part and move to Section 5.

P24, L1: There are no references of GSI, Joint Environment for Data Integration (JEDI), variational LAPS (vLAPS).

P24, L9, "We are. . .": This paragraph briefly presents an ongoing work without any evidence, results, and detailed explanations. I recommend to delete this paragraph and leave it for future papers.

P25, L18, "to assimilate observed imagery via a comparison of such with simulated imagery produced by SWIM from first guess NWP forecasts" : In my understanding, the current SWIM cannot be used for data assimilation, and the improvement is left for future works. This sentence is wrong, or it is just the authors' outlook.

P25, L33-: This and subsequent paragraphs mainly describes the author's outlook, using more spaces than pure conclusions of this paper. I think the authors have too much emphasis on the outlook.

Figure 1 presents a copy from a paper. Does this conform to the copyright? Is there no

problem to reuse the copy of the figure of Klinger et al. (2017)?

Figure 1 summarize few selected radiative transfer models, while there are significantly more 1D and 3D models over the world. The objectives of them are very different, and the purpose of presentation of this figure is not clear. It should be clarified how the models are selected. For example, are they candidates used for data assimilation in NWP, or can they all produce color imagery?

The table in Fig. 1 should be separated and presented as a table (not in a figure).

Figure 2, "...and used for subsequent calculations to estimate the radiance": tau_s should vary by location on the line of sight. How is the radiance actually calculated using tau_s and tau_o? The details should be explained in the main text possibly with equations.

---

## Referee Comment (RC2) · Anonymous Referee #2 · 5 Aug 2019

The manuscript describes an exciting visualization tool that allows comparison between output of numerical weather forecast models and camera or satellite observations. I congratulate the authors to this development! The manuscript is therefore an important contribution within the scope of the journal. However, I have major concerns about the manuscript, which need to be addressed before publication. In particular I am concerned about the frequent reference to data assimilation. For that purpose, the uncertainty of the method needs to be characterized quantitatively which hasn't been done at all. At least it is not obvious from the manuscript. My major points are, in some more detail:

[Figure]

- Description of several methods is missing, most important multiple scattering by clouds and aerosol; I was expecting some more details concerning the methods, in particular how those are applied in 3D geometry. In particular, aerosol and cloud layers observed from above are described, but I didn't find enough information about how the calculation from below and from the side is done.

- A quantitative validation of the method is missing. A number of pictures is provided which show a visual comparison of model results with camera images or satellite observations. One could infer that the model "obviously" works since the comparison looks realistic. But this type of comparison includes uncertainties of the radiation operator and differences between real situation and NWP model output; in particular for data assimilation a more quantitative characterisation of the uncertainty of the operator is required, e.g. by comparison of SWIm with independent model results.

- The manuscript contains a number of unproven claims which need to be proven in the manuscript or a reference needs to be provided; I'll give a number of examples in the specific points below.

Carefully addressing these points is critical in order to reach the scientific quality and presentation quality required for publication in AMT. To consider the second point, it would be possible to phrase the applicability to data assimilation more carefully (and to phrase it less often). Even then, however, at least some comparisons with an accurate 1D radiative transfer model are strongly suggested, which should not be that difficult.

Specific points:

Page 5, line 32: "can be benchmarked" - if so, why hasn't at least some benchmarking been done?

Page 5, line 35: Some more detail and explanation is required: I am not sure how you do this forward-backward calculation. Isn't that like a typical single scattering approach,

following the radiation from the source to the detector or the other way round?

Page 7, line 18: "integrated ... weighted by the cosine of the zenith angle"

Page 7, line 21: This is one of the above mentioned unproven claims, "the resulting radiance is nearly proportional to the spectral radiance at 540nm". Please demonstrate or provide reference!

Page 7, line 31: What you propose here is very similar to the relationship between PAR (photosynthetically active radiation) and GHI, and there is a number of references in the literature studying this relationship.

Page 8, line 7: How did you come up with these decisions/numbers?

Page 9, line 4: Please more specific! What kind of two-stream? You are interested in the radiance and thus the angular distribution - how do you handle that with the twostream? How do you apply the twostream in 3D, if you look sideways at a cloud?

Page 9, line 11: With hydrometeor density you mean the density of liquid water or ice?

Page 9, line 18: Bilinear in 3D space?

Page 9, line 24: What kind of "combination"? One in the forward and one in the backward direction, as in Key et al, Parameterization of shortwave ice cloud optical properties for various particle habits, JGR 2002?

Page 9, line 24: Why is the Henyey-Greenstein approximation needed? Couldn't you use the real phase function? Later in the text (page 21), it is stated that "SWIm was tested and can realistically reproduce rainbows, twilight sky colors and other atmospheric phenomena (Albers and Toth, 2018)." which is only possible with the real phase function but certainly not with Heyne-Greenstein.

Page 10, line 27: For an optically thick cloud one would expect $(1 + 2\mu)/3$ (old literature on asymptotic theory, and easily confirmed with a 1D radiative model). Here it is $1 + 4\mu)/3$ - how did you come up with this equation?

Page 11, line 15: The backscatter fraction is extremely important in this context - please explain in more detail and give the equations.

Page 12, line 6: Is that true? With increasing polar angle z0 the reflectance should become larger since the path length through the medium becomes longer.

Page 12, line 12: The history is not relevant here

Page 12, line 16: Faster than what?

Page 12, line 27: Equation 10 and 11 show two different expression for the double-Henyey-Greenstein function. My understanding of the DHG was more like equation 11. Which one do you use?

Page 14, line 7: Please explain in more detail what semi-empirical means in that context.

Page 16, line 5: What does that mean? Doesn't the model grid cover the whole atmosphere?

Page 17, line 3: What relatively simple functions?

Page 17, line 6: There is a well-established model of ocean BRDF by Cox and Munk, 1954. Do you use that or do you do something similar / completely different?

Page 17, line 21: I was a bit surprised that you first selected three wavelengths according to the colors RGB. Why not directly use the three wavelengths as input to the matrix? Does it make a noticable difference if the three computational wavelengths were used directly as RGB or if the described interpolation preocedure is applied?

Page 20, line 1: Validation of SWIm itself is missing (see major point above)

Page 20, line 17: As far as I understood, the method is not yet ready and the uncertainty hasn't been quantified.

Page 23, line 32: You don't have an adjoint yet, do you? Is the adjoint easily developed?

Figures are not in ascending order. E.g. Figure 9 is referenced in the text before Figure 6.

---

## Author Response (AR1)

The manuscript presents a fast, approximate 3D radiative transfer procedure for visualization of numerical weather prediction and forward modeling of ground-, aircraft-, and satellite-based camera and imager. Reflection by optically thick cloud or aerosol layer is parameterized. The scattering phase function of cloud and aerosol is approximated by the double Henyey-Greenstein function. The model is capable of produce true-color images composed of radiances at three wavelengths in the visible spectrum. While the authors emphasized the usefulness of the model in data assimilation in weather models, future works are needed to actually apply to data assimilation. Technical details of the calculation method should be clarified. I believe that important improvements are needed in the manuscript. My recommendation is therefore to make major revisions.

[Figure]

Please find below my detailed comments.

General comments

1. A lot of technical details are missing. For example, a) authors mentioned about a combination of single and multiple scattering components of radiance and some interpolation method between optically thin and thick regimes, but the methods for the combination and interpolation are not given. b) Actually, what equations are used to calculate the total radiance? It is unclear how radiances are calculated when interactions between clouds and surface reflection is present. Please show equations of total radiance that accumulates all contributions from the atmosphere, clouds, aerosols, and surface. c) The method for determining "a single scattering phase function that is equivalent to the net effect of the multiple scattering events" is missing in the current manuscript. With unknown methods, I cannot judge the validity and values of the "approximations". I have raised more points in the "specific points" below.

2. This is probably because the technical details are obscured, I cannot evaluate the model's accuracy, validity and limitations. How accurate is this model? What approximations are used? These should be clearly stated in Abstract and Conclusions.

3. As indicated in Fig. 2, the model possibly uses the bidirectional (dual-path) ray tracing along lines of sight and paths from radiation source (i.e. typically the sun) to scattering points. However, actually, it is not clear how to compute the radiance. There is no description on evaluation of the integral of scattering contributions from segments of the line of sight. Or, maybe, the model does not actually compute the integral, but instead scattering contributions from aerosol and cloud elements are approximated to be from some single point within cloud or aerosol layer. Even if so, that "point" is unknown.

4. The adjoint of this radiative transfer model is not available, and more works are needed for data assimilation. I understand the adjoint is not necessary in some frameworks. However, this paper's focus is not data assimilation, anyway. Subsection 4.3

just suggests possibilities and outlook by lengthy descriptions lacking any evidence and results. I recommend to greatly shorten this part and merge into the last paragraph of Section 5.

5. Lens flare: Lens flare effects appear in actual total-sky images, which should be discussed if comparing the model simulations with camera images. The lens flare is significant in forward scattering directions and modify the appearance of solar aureole by aerosol particles. This is critical in determining aerosol properties from total-sky images.

6. Subsection 3.2 describes the optional characteristics of the model, which is interesting. However, it is not clear whether the authors are first presenting new modeling of this complicated modeling such as moonlight, city lights, and spherical atmosphere. Are there any previous papers regarding to things presented in this section? If so, the details should be found in those papers, and this paper should just cite the references. In the current form, the descriptions are too short to fully describe the complicated radiative sources such as city light, while no results of demonstration and verification are presented in this paper. If this section should be presented in this paper, supporting information including some examples of nighttime scene and Belt of Venus as the authors mentioned in the text. Anyway, this paper does not focus on such nighttime and twilight cases. I recommend to remove this subsection and to leave them for some separate papers. I hope to see example result, validation, and applications in the future.

Specific points

Title: Does the use of acronym of "NWP" conform to the regulation of AMT?

P1, L16: In my understanding, the proposed procedure is not intended for a radiation scheme in "weather and climate" models. The first paragraph of abstract loses focus.

P5, L20, "This is a unique feature that allows. . .": The methodology is standard in the

field of atmospheric science, optics and computer vision. For example, MYSTIC, as used in Klinger et al. (2017), uses that standard method. The authors should give appropriate references.

P6: Eqs. (1) and (2) are identical, and there is no reason to present both.

P6, L6, Fig. 2 and Table 1: A method for radiance integration is not clear. In my understanding, the radiance should be an integral of contributions coming from many small segment of line of sight (from camera). Each contribution is a function of irradiance from radiative source, which should be calculated by the forward path using some spatial interpolation or directly tracing a ray from the scattering point to the radiative source. This kind of integration should be explained in detail. Without such a description, the readers cannot understand the relationship between forward and backward rays.

P6, L22, "615nm...": Please give a reason for selection of these 3 wavelengths. Is there any references?

P7, L1, "The light...": The same sentence is given previously.

P7: Eqs. (3) and (4) are identical, and there is no reason to present both.

P9, L4, "A two-stream approach is used to incorporate the backscatter fraction...": This sentence is not clear. There are several two-stream approaches. References should be given. Or, is this paper presenting a new two-stream approach for rendering? The definition of backscatter fraction should be clarified. How is the backward fraction used to determine the total downward illumination? What is "illumination"? Is it equal to the irradiance?

P9, L24: Why are HG functions used? Why not Mie theory? The reason should be explained in the text.

P10, Eq. (6): Explanations of $i$ and theta are missing.

P10, Eq. (7): An explanation of $c_i$ is missing.

[Figure]

P10, L4: "When $\tau \ll 1$": Please give an exact definition of tau. In Eq. (2), tau is optical depth integrated along the beam path. The definition of "beam path" is not clear because there are two paths: from camera to target and from the sun to target. In Eq. (8), how is tau defined? Is it the optical thickness along line of sight, along the direct source beam, or some combination of them? For volumetric object (i.e. cloud), there are infinite number of forward paths (from sun) corresponding to small path segments along the backward path (from camera).

P10, L15-17, "more efficient approximation that arrives at a single scattering phase function...": This approximated phase function is not given in Subsubsection 3.4.2. I also failed to find any description in the manuscript. I guess the phase function should be modified to represent multiple scattering effects. How are they modified from the original single-scattering phase function?

P10, L17-18: The interpolation scheme is not actually given in Subsubsection 3.4.2. I also failed to find any details of this interpolation in the manuscript. What interpolation scheme is actually used between optically thin and thick clouds? Please show it using equations if possible.

P10, L26: Does "heavy overcast sky" mean sky with 100% cloud cover? How is the radiance parameterized as a function of cloud cover? How is the radiance computed when the sky is not overcast?

P11, L7, "Intermediate values of tau_0 are given empirical phase functions": How are tau_0 and the empirical phase functions related?

P11, L28, "A simple bidirectional...": How is the BRDF developed? Is it based on any measurements or theoretical calculations using rigorous radiative transfer models? It is better to show equations of the ARF.

P12, L9: Actually, what and how is interpolated? Is the cloud albedo (or BRDF) inter-polated? Is it linear interpolation with respect to tau? It is better to explain the method

with equations and references (if present).

P13, L25: This paragraph mentioned about single-scattering albedo (for single scattering radiance), but previous explanations say that scattering contributions by aerosols do not depend on the single-scattering albedo (Eq. (8)). In P10, L7, single-scattering radiance is insensitive to the single-scattering albedo: "This relationship applies to hydrometeors as well as aerosols". If so, why single-scattering albedo is discussed in P13.

P14, L6, "The first row in Table 2 was derived semi–empirically for relatively dusty days": How was the camera radiometrically calibrated? Were all pixels of camera image used? Was the circumsolar region excluded or obscured by shadow band or anything? Of course, saturated pixels should be excluded. Are there any references?

P14, L11, "AERONET": References are needed.

P15, L3-7, "As with cloud multiple scattering, a rigorous approach such as Monte–Carlo would consider each scattering event explicitly, though this would be computationally inefficient. . . . multiple scattering events.": These 2 sentences are almost identical as previously shown in the hydrometeor subsection. Please make a point clear.

P15, L10 and L21, "eq. 6": The equation number seems wrong.

P15, L11: What are CIRA, RAMS, and WRF? They are explained at page 18, but it is too late. Also, references are needed.

P16, L6, "these quantities are merged together to provide the combined radiance": How are they merged? Is the combination simply a sum of radiance components by clear sky and aerosol/cloud? How does the model implement the transmissivities between the sun and cloud/aerosol and between the cloud/aerosol and the camera? Please show equations if possible.

P17, L2-5: A lot of details and references are missing. What is "relatively simple an-

alytical function"? How is the f "modified"? What a sun glint model is chosen from many previously proposed models? How is "scattering from below the water surface" modeled? If there are references, please cite them. If there is no reference, described the details.

P18, L3, "3x3 transfer matrix": Please provide this matrix explicitly or show a reference. This is because there are several variants of RGB color space.

P19, L2, "A more complete…": This sentence is not clear. Is this confirmed by tests or just a guess by the authors? Is there any evidence?

P18, L31, "we've" → "we have"

P18, L31: "Simulated Weather Imagery (SWIm)" → "SWIm" (This is already written at P4, L7 and should not be repeated)

P20, L23, "Local Analysis and Prediction System Â∎ LAPS, Toth et al., 2014)" This should be presented earlier because LAPS appears at P19, L9.

P20, L25, "METAR": What is METAR?

P21, L8, "High Resolution Rapid Refresh (HRRR)" → "HRRR". HRRR first appeared earlier. It should be explained when presented first.

P21, Subsubsection 4.2.1: In this comparison of simulated observed imagery, strong direct sun beam can cause lens flare, which should be taken into account. The lens flare may be significant particularly when strong direct beam is incident to the camera as in Fig. 11. The camera image cannot be directly compared with the simulated image.

P21, L26, "The brightness scaling...": This should be primarily influenced by the lens flare.

P22, L7, "GFS": What is GFS?

[Figure]

P22, L17, "uncertainty in the brightness scaling of the DSCOVR imagery" : Why is it uncertain? If the radiance data are from the EPIC product, the brightness scaling should be the same as for simulated imagery.

P23, L32, "One approach would entail developing SWIM's Jacobian or adjoint, while other techniques employ recursive minimization.": This sentence is not clear. Please give references. The authors stated that "SWIm can be used as a forward operator in a variational minimization". However, the derivatives are required in the variational minimization, while derivative calculations are not in scope of the current manuscript. This is confusing. It is not very clear whether he current SWIM be used for data assimilations or further works are needed.

P23, Subsection 4.3: This section seems to be the authors' outlook. The problem is that it is not clear whether it is outlook or not. It is better to shorten this part and move to Section 5.

P24, L1: There are no references of GSI, Joint Environment for Data Integration (JEDI), variational LAPS (vLAPS).

P24, L9, "We are...": This paragraph briefly presents an ongoing work without any evidence, results, and detailed explanations. I recommend to delete this paragraph and leave it for future papers.

P25, L18, "to assimilate observed imagery via a comparison of such with simulated imagery produced by SWIM from first guess NWP forecasts" : In my understanding, the current SWIM cannot be used for data assimilation, and the improvement is left for future works. This sentence is wrong, or it is just the authors' outlook.

P25, L33-: This and subsequent paragraphs mainly describes the author's outlook, using more spaces than pure conclusions of this paper. I think the authors have too much emphasis on the outlook.

Figure 1 presents a copy from a paper. Does this conform to the copyright? Is there no

[Figure]

problem to reuse the copy of the figure of Klinger et al. (2017)?

Figure 1 summarize few selected radiative transfer models, while there are significantly more 1D and 3D models over the world. The objectives of them are very different, and the purpose of presentation of this figure is not clear. It should be clarified how the models are selected. For example, are they candidates used for data assimilation in NWP, or can they all produce color imagery?

The table in Fig. 1 should be separated and presented as a table (not in a figure).

Figure 2, "…and used for subsequent calculations to estimate the radiance": tau_s should vary by location on the line of sight. How is the radiance actually calculated using tau_s and tau_o? The details should be explained in the main text possibly with equations.
* * *
[Figure]

Atmos. Meas. Tech. Discuss.,
doi:10.5194/amt-2019-81-RC2, 2019

[Figure]

The manuscript describes an exciting visualization tool that allows comparison between output of numerical weather forecast models and camera or satellite observations. I congratulate the authors to this development! The manuscript is therefore an important contribution within the scope of the journal. However, I have major concerns about the manuscript, which need to be addressed before publication. In particular I am concerned about the frequent reference to data assimilation. For that purpose, the uncertainty of the method needs to be characterized quantitatively which hasn't been done at all. At least it is not obvious from the manuscript. My major points are, in some more detail:

[Figure]

- Description of several methods is missing, most important multiple scattering by clouds and aerosol; I was expecting some more details concerning the methods, in particular how those are applied in 3D geometry. In particular, aerosol and cloud layers observed from above are described, but I didn't find enough information about how the calculation from below and from the side is done.

- A quantitative validation of the method is missing. A number of pictures is provided which show a visual comparison of model results with camera images or satellite observations. One could infer that the model "obviously" works since the comparison looks realistic. But this type of comparison includes uncertainties of the radiation operator and differences between real situation and NWP model output; in particular for data assimilation a more quantitative characterisation of the uncertainty of the operator is required, e.g. by comparison of SWIm with independent model results.

- The manuscript contains a number of unproven claims which need to be proven in the manuscript or a reference needs to be provided; I'll give a number of examples in the specific points below.

Carefully addressing these points is critical in order to reach the scientific quality and presentation quality required for publication in AMT. To consider the second point, it would be possible to phrase the applicability to data assimilation more carefully (and to phrase it less often). Even then, however, at least some comparisons with an accurate 1D radiative transfer model are strongly suggested, which should not be that difficult.

Specific points:

Page 5, line 32: "can be benchmarked" - if so, why hasn't at least some benchmarking been done?

Page 5, line 35: Some more detail and explanation is required: I am not sure how you do this forward-backward calculation. Isn't that like a typical single scattering approach,

following the radiation from the source to the detector or the other way round?

Page 7, line 18: "integrated ... weighted by the cosine of the zenith angle"

Page 7, line 21: This is one of the above mentioned unproven claims, "the resulting radiance is nearly proportional to the spectral radiance at 540nm". Please demonstrate or provide reference!

Page 7, line 31: What you propose here is very similar to the relationship between PAR (photosynthetically active radiation) and GHI, and there is a number of references in the literature studying this relationship.

Page 8, line 7: How did you come up with these decisions/numbers?

Page 9, line 4: Please more specific! What kind of two-stream? You are interested in the radiance and thus the angular distribution - how do you handle that with the twostream? How do you apply the twostream in 3D, if you look sideways at a cloud?

Page 9, line 11: With hydrometeor density you mean the density of liquid water or ice?

Page 9, line 18: Bilinear in 3D space?

Page 9, line 24: What kind of "combination"? One in the forward and one in the backward direction, as in Key et al, Parameterization of shortwave ice cloud optical properties for various particle habits, JGR 2002?

Page 9, line 24: Why is the Henyey-Greenstein approximation needed? Couldn't you use the real phase function? Later in the text (page 21), it is stated that "SWIm was tested and can realistically reproduce rainbows, twilight sky colors and other atmospheric phenomena (Albers and Toth, 2018)." which is only possible with the real phase function but certainly not with Heyne-Greenstein.

Page 10, line 27: For an optically thick cloud one would expect $(1 + 2\mu)/3$ (old literature on asymptotic theory, and easily confirmed with a 1D radiative model). Here it is $1 + 4\mu)/3$ - how did you come up with this equation?

[Figure]

Page 11, line 15: The backscatter fraction is extremely important in this context - please explain in more detail and give the equations.

Page 12, line 6: Is that true? With increasing polar angle z0 the reflectance should become larger since the path length through the medium becomes longer.

Page 12, line 12: The history is not relevant here

Page 12, line 16: Faster than what?

Page 12, line 27: Equation 10 and 11 show two different expression for the double-Henyey-Greenstein function. My understanding of the DHG was more like equation 11. Which one do you use?

Page 14, line 7: Please explain in more detail what semi-empirical means in that context.

Page 16, line 5: What does that mean? Doesn't the model grid cover the whole atmosphere?

Page 17, line 3: What relatively simple functions?

Page 17, line 6: There is a well-established model of ocean BRDF by Cox and Munk, 1954. Do you use that or do you do something similar / completely different?

Page 17, line 21: I was a bit surprised that you first selected three wavelengths according to the colors RGB. Why not directly use the three wavelengths as input to the matrix? Does it make a noticable difference if the three computational wavelengths were used directly as RGB or if the described interpolation preocedure is applied?

Page 20, line 1: Validation of SWIm itself is missing (see major point above)

Page 20, line 17: As far as I understood, the method is not yet ready and the uncertainty hasn't been quantified.

Page 23, line 32: You don't have an adjoint yet, do you? Is the adjoint easily developed?

[Figure]

Figures are not in ascending order. E.g. Figure 9 is referenced in the text before Figure
6.

[Figure]

Reviewer #1

General Comments:
* * *
1. More details on the forward light rays and 3D irradiance field are now discussed in Section 3.4.

a) the method of interpolating between the single & multiple scattering cases is now given in section 3.4.2

b) The final combination of clear and cloudy contributions to the radiance is now specified in section 3.6, with the individual steps described in earlier sections. Surface albedo is discussed in section 3.7.

c) the procedure for handling multiple scattering with an effective single scattering phase function was elaborated upon for cloud liquid. Similar formulations (not shown) are used for cloud ice, rain and snow.

2. Additional equations have been added in various sections to more completely describe the total (and solar relative) radiance. The overall accuracy of SWIm is now summarized in Table 4, and the items within this table are discussed in the conclusion and elsewhere in the manuscript.

3. Computation of radiance is now given in greater detail throughout the manuscript (e.g. section 3.4.1).

4. We believe section 4.3 gives a useful review of related 3D assimilation methods that include the use of visible light wavelengths and cameras. We now provide some results (Figure 14) that illustrate preliminary steps we are taking to develop a SWIm based assimilation. We agree there is much more to be done.

5. Lens flare is now mentioned in a more general context at the end of section 4.2.3.

6. Details on moonlight, city lights, and spherical atmosphere will be deferred to a future paper and this has been clarified in the text.

Specific points
* * *
Title: "NWP" is now spelled out

P1, L16: The first paragraph of the abstract provides context about the importance of visible wavelength radiation in modeling and we believe the 3rd paragraph of the abstract discusses the more focused role of SWIm in a reasonable manner.

P5, L20: The last paragraph of section 2 along with parts of section 3.8 have been modified to reflect the reviewer's suggestions.

P6, Eqs. (1) and (2) are indeed identical. We now only show the second equation.

P6, L6: The method for radiance integration, with Step 2 (clear sky) ray tracing as an example, is now shown in Eq. 3.

P6, L22:  The rationale for wavelength selection was elaborated upon.

P7,  L1:  The redundant sentence was removed

P7, Eqs. (3) and (4) are indeed identical. We now only show the second equation.

P9, L4: This line and section has been revised to improve clarity. "Two-stream" isn't mentioned now since we use a different relatively simple approach. "Illumination" has also been replaced by "irradiance" in this section.

P9, L24: Rationale for using HG functions is now given in sections 3.4.1 and 3.4.2

P10, Eq 6: Definitions of i and theta are now in place

P10, Eq 7: c(i) was changed to f(i) and is now defined as a summation

P10, L4: Tau is now more clearly defined in this context

P10, L15-17, 17-18: The phase function is now more completely described for the case of cloud liquid, though the formulation isn't yet detailed for cloud ice, rain, or snow.

P10 L26: In the context of this section, "heavy overcast" means the 3D irradiance field (eq. 8) at the location of the portions of the cloud along the line of sight closest to the observer is ~<0.4. This definition of overcast is independent of cloud fraction and related quantities.

P11 L7: The intermediate phase functions are now described in section 3.4.2.
P11 L28: A simple ARF parameterization was developed with  references and equations now given in section 3.4.3.

P12, L9: It is now stated that linear interpolation with respect to cloud albedo is used to approximate the reflectance between the low $\tau$ and high $\tau$ regimes.

P13 L25: Single scattering albedo is now included in eq. 11 since this is considered for aerosols with the single scattering radiance calculation.

P14, L6: The items mentioned by the reviewer are now clarified in the text within section 3.5.2.

P 14, L11: An AERONET reference was added in section 3.5.2.

P15, L3-7: These two sentences are now condensed for clarity and to avoid repetition.

P15, L11: These chemistry models are now better explained here, including references.

P16, L6: More details and equations are now given in section 3.6.

P17, L2-5: A reference was added that we base the ocean reflectance upon. Brief descriptions are given for the handling of land anisotropic reflectance.

P18, L3: The transfer matrix is now explicitly supplied

P18, L31: Wording adjusted to follow both suggestions

P19, L2: "A more complete" appears in the text on P18, L25. A more specific reference to Rayleigh correction is being added for the satellite example. For everyday photography this is more of a general comment that images often have more saturation, or may suppress the atmospheric brightness with polarizing filters and the like, all for the purpose of making the image look more appealing.

P20, L23: The LAPS reference was moved to section 4.1 and the LAPS description was clarified in section 4.2.

P20, L25: A more general description is now used for the METAR observations

P21, L8: The HRRR acronym is now expanded upon its first use.

P21, sec 4.2.1: We agree and lens flare is now mentioned there in the text.

P21, L26: In a camera image, the regions that are saturated (hence not useful for quantitative brightness comparison) can reach that brightness from either lens flare or sunlight scattering by aerosols and clouds, depending on the situation and quality of the camera. A clarifying sentence was added here.

P22, L7: GFS is now defined

P22, L17: The strategy for producing figure 12 is now explained in more detail to address the reviewer questions, within section 4.2.2.

P23, L32: We state that "One approach would entail developing SWIm's Jacobian or adjoint". This should clearly imply that it has yet to be done.

P24, L1: References for vLAPS, GSI were added. JEDI is now referenced with a website since this appears to be unpublished at this time.

P25, L18: The revised text now describes a simple camera assimilation technique we performed that can serve as an introduction to the other methods mentioned in our roadmap.

P25, L33: As now mentioned in the last paragraph of Section 1, this study is intended to introduce SWIm, and describe what has been done so far, and suggest a roadmap for the future.

Figure 1: We are no longer using the image from another paper. This has now been converted into Table 1. The table is intended to show a variety of RT packages and for context to illustrate which ones have similar capabilities as SWIm. The other questions are now addressed in section 2 of the text.

Figure 2: Section 3.3 now has an equation illustrating how radiance is computed as an integral from the rays traced in the figure. In this section, the simple example of Rayleigh or Mie single scattering is illustrated. Additional equations relating to multiple scattering are given in section 3.4.

Reviewer #2

General Comments:
* * *
The authors are glad the reviewer appreciates the relevance and value of SWIm. We appreciate the major concerns and would like to respond both generally and specifically. Data assimilation is now mentioned mainly in the context of future work. However even with significant approximations in the radiative transfer it is possible to perform simple types of assimilation with metrics like the correlation coefficient as now described in the text. Evaluation of the 1D radiative transfer has been performed in the context of the distribution of reflectance values at the red wavelength in DSCOVR / EPIC imagery for both clear and cloudy regions. Oceanic clear areas are in the expected range of 5-6% reflectance factor with the bright tops of tropical convective clouds between 1.0 and 1.1.

Specific points
* * *
P 5, L32: As now discussed in Section 5, specific comparisons with other radiative transfer packages (e.g. CRTM, MYSTIC) is a good topic for future work. Thus far we've focused mainly on comparisons with ground-based cameras, pyranometers, and DSCOVR imagery, even though they typically include the LAPS cloud analysis used for SWIm input in the evaluation pipeline.

P 5, L35: The forward-backward ray-tracing procedure has now been clarified in section 3 of the text.

P 7, L18: The zenith angle weighting is now mentioned in the text.

P 7, L21: A simple calculation of this was performed and now summarized in the last paragraph of section 3.1.

P 7, L31: We would like to investigate this further in the literature and report in a followup paper. Thus far the authors have only seen information about this in the form of an online solar spectrum calculator used within the solar power industry.

P 8, L 7: The statements about the moon's brightness came from a literature search about the "opposition effect". It is now made clear in the manuscript that details about the moon's brightness will be deferred to a future paper.

P 9, L 4: The "two-stream" term was used too loosely and the description has now been updated to make the detailed procedure more clear.

P 9, L11: The density is based on the hydrometeor type and the effective radius as now mentioned in the text.

P 9, L18: "bilinear" is now replaced with "trilinear" since light rays are traced in 3D space.

P9, L24: The linear combination of HG functions is now introduced in section 3.4.1 and further described in section 3.4.2 and Appendix B. The HG function terms provide for both forward and backward scattering.

P9, L24: Rationale for using HG functions is now given in sections 3.4.1 and 3.4.2, particularly with the convenience of being able to raise "g" to an exponent to approximate multiple scattering.

P 10, L 27: This was chosen empirically, partly since it averages to 1 with respect to zenith angle. We will try your formulation since it will probably help improve the pyranometer comparisons with overcast conditions, camera image comparisons with partly cloudy conditions, and have better theoretical footing as you suggest.

P11, L15: The procedure for calculating the backscatter fraction is now given in section 3.4.3.

P12, L6: We would suggest the increasing optical path of the sunlight through optically thin cloud or aerosols shouldn't affect the observed radiance (technically the reflectance factor), since we are in a single scattering regime. The path length from the observer through the medium is remaining constant.

P12, L12: The HAALE-MURI history has been removed, while this project is represented in the Acknowledgements section.

P12, L16: 1-D aerosol calculations are faster than 3-D aerosols as now clarified in the text.

P12, L27: Eq. 11 represents a pair of DHG functions from eq. 10 as now explained further in the text.

P14, L7: The semi-empirical procedure is now explained in more detail in the text.

P16, L5: SWIm is designed to work even in cases when the NWP grid is limited in horizontal or vertical extent. This helps save computing resources and allows SWIm to work with limitations in NWP systems.

P17, L3: A reference was added that we base the ocean reflectance upon - this is the same one the reviewer suggested.

P17, L6: A reference was added that we base the ocean reflectance upon. Brief descriptions are given for the handling of land anisotropic reflectance.

P17, L21: The Bell et al. reference describes some experiments that help show the value of having a more complete spectrum to get the best chromaticities and color rendering. The interpolation procedure we describe will by design produce a more accurate spectrum and hence chromaticity, compared with simply inserting three narrowband wavelengths into the CIE color matching functions. We also in the text now give the rationale for selecting the three reference wavelengths used within SWIm.

P20, L1: In addition to image correlation, subjective evaluation of the 1D radiative transfer has been performed in the context of the distribution of reflectance values at the red wavelength in DSCOVR / EPIC imagery for both clear and cloudy regions. Ground-based comparisons of global horizontal irradiance (GHI) have also been done. A more rigorous comparison of SWIm with another 3D radiative transfer model (e.g. MYSTIC, SHDOM) is planned for a future paper..

P20, L17: The solar irradiance (GHI) comparisons are now being done with case studies of clear and partly cloudy conditions (e.g. section 3.1 - Fig. 2, section 4.2.1 - Figs 2,10), and overcast skies though not yet in a more systematic manner.

P23, L32: We have clarified in the text that the adjoint has yet to be developed. We think it is feasible to do in the future. Minimization methods that do not require an adjoint would also be possible.

[revised manuscript text omitted]
 ~~radiance at 540nm. The proportionality holds for the solar spectrum, and typical modifications of this spectrum resulting from Rayleigh and Mie scattering. For example a normalized Rayleigh spectrum has more intensity in the blue wavelengths and less in the red and IR compared with the solar spectrum. The break even point is close to 540nm, and happens to be close to the 550nm standard often used to represent the peak sensitivity in human vision. This is borne out in preliminary case studies~~irradiance at the 546nm green wavelength used in SWIm calculations. This approximation is reasonably accurate in cases where the global irradiance has a similar spectrum to the incident solar radiation, as seen on a mostly cloud-free day in Fig. 2. For example the slight reddening of the direct solar radiation due to Rayleigh scattering is often partially compensated by the blue color of the sky that represents the diffuse irradiance. Overcast sky conditions should work as well as long as the sky is a relatively neutral

gray color. Indeed, the existing algorithm generally provides a close match when comparing SWIm generated GHI values to actual GHI values measured with a pyranometer at the National Renewable Energy Laboratory (NREL), in Golden, CO. We are presently working to refine this algorithm with a correction parameter based on atmospheric water vapor content, since this does have a more selective effect in the near-IR wavelengths. Similar calculations can be made for direct and diffuse solar irradiance.

**3.2 Moonlight, city lights, and other light sources**

During the night the moon can replace the sun as an angularly localized light source (e.g., Miller et al., 2009). In the present study, the lunar radiance is calculated from considerations of its astronomical magnitude as a function of phase angle (180° – scattering angle). Near full moon a correction is added based on the opposition effect and potential lunar eclipses. At phase angles of less than ~4° the brightness is increased by up to ~20%, except the brightness is reduced substantially to factor in lunar eclipses as we move closer to 0° phase angle. Near new moon a term for Earthshine is taken into account, since this becomes significant compared with reflected sunlight from the lunar crescent. Earthshine is sunlight reflected first from the Earth's surface, then from the moon's surface.

Other light sources such as city lights, airglow, zodiacal light, individual stars and galactic glow are also included. An approximate scattering calculation is performed for city lights emanating from spatially extended areas with respect to the gas, aerosols, and cloud components of the atmosphere.

Since the Earth can be approximated as a spherical object, various twilight phenomena can be displayed via spherical geometry accommodated by SWIm. The varying path of light through the curved atmosphere enables the reproduction of observable optical effects, including changes in clear sky and cloud colors. Effects relating to the Earth's shadow (including blockage by the terrain and attenuation by the lower atmosphere) are also represented, affecting both the molecular atmosphere and cloud related radiative processes. The "Belt of Venus" can be simulated when a moderate amount of high-altitude aerosols scatter red light just above the Earth's shadow, except it tends to overestimate the GHI in uniform overcast conditions. We are considering whether this is due to the radiative transfer assumptions in SWIm or an underestimation in the analyzed 3D hydrometeors and associated cloud optical thickness.

In a worst case scenario of a pure Rayleigh blue sky, we calculate that the normalized spectrum integrated from 0.3 $\mu$ to 3.0 $\mu$ has a crossover point at 530nm with the solar spectrum, yielding an irradiance underestimation of about 11% of the diffuse component when a SWIm reference green wavelength of 546nm is used. With a high sun in a clear sky this reduces to about 1% total GHI error since the Rayleigh scattered diffuse component is a small proportion of the total irradiance. For this error estimation, we integrated the Planck function at 5800K to represent an approximate solar spectrum and compared this with the Planck function convolved with the $\lambda^{-4}$ intensity vs wavelength associated with Rayleigh scattering. The error be reduced by a more detailed consideration of the three SWIm reference wavelengths. A simple preliminary correction parameter based on atmospheric water vapor content has been added to account for absorption in the near-IR wavelengths. This presently neglects separate consideration of direct and diffuse solar irradiance.

**3.2 Other light sources and atmospheric effects**

With its realistic 3D ray tracing, SWIm is able to simulate a number of daytime, twilight, and nighttime atmospheric light effects, including consideration of a spherical atmosphere. This involves various light sources including moonlight, city lights, airglow, and astronomical objects. These will be demonstrated in a separate paper.

**3.3 Clear sky ray-tracing**

To cover the full extent of atmosphere beyond the NWP model domain, a "clear sky" ray-tracing (Step 2) is conducted  on a coarser angular grid compared with Step 1. The primary purpose of Step 2 is to provide a more direct account of the radiance produced by Rayleigh single scattering. A second purpose is to model the effect of aerosols that may extend beyond the top of the model grid, specified via a 1-D stratospheric variable. The accuracy of radiative processes associated both with stratospheric aerosols and twilight  benefit from the vertical extent considered in this step, all the way up to about 100km. To calculate the solar relative spectral radiance, the ray-tracing algorithm integrates along each line of sight from the observer as

$$L'_{\lambda,clear} = P(\theta) \int e^{-\tau_s} e^{-\tau_o} d\tau_o \qquad (3)$$

where $\theta$ is the scattering angle shown in Fig. 1 and $P(\theta)$ is the phase function (described in section 3.4.1). $\tau_s$ is the optical thickness along the forward ray (yellow

lines in Fig. 1) between the light source and each point of scattering and $\tau_o$ is the optical thickness along the backward ray (purple lines in Fig. 1) between the observer and each scattering point. We will denote this to be the clear sky radiance, that includes the molecular component through the full atmospheric depth and aerosols above the model grid top.

**3.4 Hydrometeors**

As the light rays are traced through the model grid (yellow rays in Fig . 1, Step 1a in Table 2) their attenuation and forward scattering is determined by considering the optical thickness of intervening clouds and aerosols along their paths. The optical thickness between each 3D grid point and the light source $\tau_s$ is calculated. An estimate of back-scatter fraction $b$ is incorporated to help determine the scalar irradiance $E_\lambda$ (direct +  scattered) at a particular model grid point. $b$ is assigned a value of $.063$ for cloud liquid and rain, $.14$ for cloud ice and snow, and $.125$ for aerosols. Scalar irradiance is the total energy per unit area impinging on a small spherical detector. Based on a cloud radiative transfer parameterization (Stephens, 1978), a simplified version was developed for each 3D grid point as follows,

$$T_1 = 1 - \frac{b\tau_s}{(1+b\tau_s)} \tag{4}$$

[revised manuscript text omitted]
 can be used by data assimilation and model developers as a validation to assess model performance and help guide improvements in initial and forecast fields, respectively. The imagery provides a qualitative validation of both the model fields and the visualization package when simulated images are compared against actual camera images. If in various situations simulated imagery can well reproduce observed images, this is an indication of the realism of the radiative transfer / visualization package (i.e., SWIm). Discrepancies between simulated and observed images in other cases may be interpreted as shortcomings in the analyzed or model forecast states. ¶
¶
The vantage point for such assessments can be from a variety of vantage points (on the ground, in the air, or fromin space, (i.e. with multi-spectral visible satellite data). At a glance various obstructions to visibility can be intuitively seen in the imagery such as clouds, haze, and smoke. The land surface state including snow cover, visibility and illumination can be assessed. Figure 4.3 shows a cloud-free sky comparison where aerosol loading was relatively high due to smoke. ¶
¶
Solar irradiance computed by a solid angle integration of SWIm imagery can be compared with corresponding pyranometer measurements. For space-based satellite imagery, color images can be compared qualitatively and visible band reflectance can be used for quantitative comparison.¶

¶

SWIm has been tested with a variety of models. An example is an NWP system that produces very rapid update (5-15min) and very high resolution (e.g. 500m) analyses and forecasts (Local Analysis and Prediction System - LAPS, Toth et al., 2014). The cloud analysis (Albers et al., 1996, Jiang et al., 2015) of LAPS uses satellite (including IR and 1-km resolution visible imagery, updated every 15-min), METAR, radar, and aircraft observations along with a first guess forecast to produce 3-D fields of cloud and hydrometeor variables. The LAPS cloud analysis is running regionally at 500m horizontal resolution on a 5- to 10-min update cycle. The 3-D hydrometeor fields are analyzed using satellite, radar, surface ceilometer observations, and model first guess fields. The largely sequential data insertion procedure of today's LAPS 
[revised manuscript text omitted]

| Multiple Scattering | Approximate | Y | Y | Y | Y |
| Fast Running | Y | Y | Y | N | N |
| Ground- air- or space-based observer | All | Space | Space | All | All |
| Curved Earth Shadow / Twilight | Y | N | N | | Y |
| Moon/Stars/City Lights | Y | N | N | | |
| 2-D (directional) images | Y | Y | TOA SW up (Isotropic) | Y | Y |
| Wavelengths | VIS | VIS + IR | VIS + IR | | |
| Grid Resolutions | All | All | All | <=100m | All |

Gold Standard

(Klinger, Mayer et al., 2017)

Figure 2.1. Overview showing features of interest for a sampling of radiative transfer packages.

[Figure]

**θ = scattering angle**

**τ = cloud / aerosol optical thickness**

[Figure]

Figure 3.11. General ray-tracing procedure showing forward light rays (yellow) coming from the light source. A second set of light rays (pink) are traced backward from the observer. The forward and backward optical thicknesses ($\tau_s$ and $\tau_o$ ) are calculated along these lines of sight and used for subsequent calculations to estimate the radiance on an angular grid as seen by the observer.

[Figure]

Figure 2. Time series of GHI values integrated from SWIm radiance images (red lines, vertical axis on left) compared with concurrent pyranometer observations in $Wm^{-2}$ at NREL (green lines). The comparison spans a 4 hour period on the morning of August 12, 2019. Simulated minus pyranometer GHI values are plotted as blue circles (vertical axis on right). Sky conditions were free of significant clouds, with aerosol optical depth < 0.1.

[Figure]

- 0° scattering angle looks toward the sun, 180° is opposite
- Phase function (y-axis) averages to 1 over the sphere
- Approximation to more rigorous Mie theory
- Multi-parameter Henyey-Greenstein functions
  - each term uses asymmetry factor "g"

[Figure]

- 0° scattering angle looks toward the sun, 180° is opposite
- Phase function (y-axis) averages to 1 over the sphere
- Approximation to more rigorous Mie theory
- Multi-parameter Henyey-Greenstein functions
  - each term uses asymmetry factor "g"

Figure 3.23. Single scattering phase functions used for cloud liquid, cloud ice, rain, and snow.

**Aerosol Phase Functions**

[Figure]

[Figure]

[Figure]

[Figure]

Figure 3.34. Simulated panoramic images with an AOD of 0.1 using the Colorado empirical phase function (a), and the Mie theory mixed dust case (b). These two phase functions are compared in (c).

[Figure]

Figure 3.4. View from space of the NAVGEM global model, using aerosols only. The perspective point is $1.5 \times 10^6 km$ distant.

[Figure]

Figure 3.55. Simulated image of a HRRR-Smoke forecast with a smoke plume from the December 2017 California wildfires. The view is zoomed in from a perspective point at $40000\ km$ altitude.

[Figure]

Figure 3.66. View from space of the NAVGEM global model, using aerosols only. The perspective point is $1.5 \times 10^6 km$ distant.

[Figure]

Figure 7. In-situ panoramic view in the lower troposphere showing smoke aerosols and hydrometeors. This is part of an animation simulating an airplane landing at the Denver International Airport. The panorama spans $360^o$ from a perspective $\sim 4km$ above ground. Hydrometeor fields are from a LAPS analysis.

[Figure]

[Figure]

Figure 3.78. SWIm generated image for a hypothetical clear-sky case having an aerosol optical depth ~0.05. The model grid and associated terrain data is at 30m resolution and surface spectral albedo information is derived from 0.7m resolution aerial imagery from the USDA. The vantage point is from the U.S. Department of Commerce campus in Boulder, Colorado, looking at azimuths from south through west.

[Figure]

[Figure]

Figure . View  from  *4km (a) and 20m* (b) above the Persian Gulf of a RAMS model simulation showing dust, hydrometeors, land surface, and water including sun glint, displayed with a cylindrical (panoramic) projection.

[Figure]

[Figure]

Figure 4.210. Comparison of observed  (right) to simulated (left)  polar equidistant projection images showing the upward looking hemisphere from a ground-based location in Golden, Colorado on September 27, 2018 at 2250UTC. LAPS analysis fields are used for the simulated images.

[Figure]

Figure 11. A comparison of aerosols at 2100UTC on August 20, 2018 in Golden, Colorado showing  a panoramic simulated  (top) and an all-sky camera image (bottom). The correlation $\bar{r}$ between the images is denoted as 0.961.

[Figure]

Figure 4.412. Side-by-side comparison (SWIm image on the left, DSCOVR-EPIC image on the right). Both images are from approximately 1800UTC on April 28, 2019.

[revised manuscript text omitted]